# Mind the Gap: Bridging Thought Leap for Improved Chain-of-Thought Tuning

**Haolei Xu**[1][*]   **Yuchen Yan**[1][*]   **Yongliang Shen**[1][†]   **Wenqi Zhang**[1]   **Guiyang Hou**[1]
**Shengpei Jiang**[2]   **Kaitao Song**[3]   **Weiming Lu**[1][†]   **Jun Xiao**[1]   **Yueting Zhuang**[1]

[1] Zhejiang University   [2] SF Technology
[3] Microsoft Research Asia
{xuhaolei,syl,luwm}@zju.edu.cn

Project: https://zju-real.github.io/CoT-Bridge

## Abstract

Large language models (LLMs) have achieved remarkable progress on mathematical tasks through Chain-of-Thought (CoT) reasoning. However, existing mathematical CoT datasets often suffer from **Thought Leaps** due to experts omitting intermediate steps, which negatively impacts model learning and generalization. We propose the CoT Thought Leap Bridge Task, which aims to automatically detect leaps and generate missing intermediate reasoning steps to restore the completeness and coherence of CoT. To facilitate this, we constructed a specialized training dataset called **ScaleQM+**, based on the structured ScaleQuestMath dataset, and trained **CoT-Bridge** to bridge thought leaps. Through comprehensive experiments on mathematical reasoning benchmarks, we demonstrate that models fine-tuned on bridged datasets consistently outperform those trained on original datasets, with improvements of up to +5.87% on NuminaMath. Our approach effectively enhances distilled data (+3.02%) and provides better starting points for reinforcement learning (+3.1%), functioning as a plug-and-play module compatible with existing optimization techniques. Furthermore, CoT-Bridge demonstrates improved generalization to out-of-domain logical reasoning tasks, confirming that enhancing reasoning completeness yields broadly applicable benefits.

## 1   Introduction

Large Language Models (LLMs) [1–6] have demonstrated significant performance improvements through Chain-of-Thought (CoT) reasoning [7, 8], particularly on complex tasks [9] such as mathematics [10] and coding. CoT guides models to generate intermediate reasoning steps, simulating the human process of step-by-step problem solving, thereby enhancing both solution accuracy and interpretability [11]. The quality of these reasoning chains directly impacts model performance, serving as a critical foundation for advanced reasoning capabilities.

Despite substantial progress, we identify a prevalent but understudied phenomenon in CoT datasets [12–16]: **Thought Leap**. This refers to instances where one or more intermediate reasoning steps are omitted between adjacent steps, creating cognitive gaps in the reasoning chain. Unlike factual errors or answer inaccuracies that have been extensively studied, Thought Leap specifically concerns the *completeness* of reasoning structures. Figure 1 (a) illustrates this phenomenon through a mathematical problem where critical bridging steps (highlighted in gold) are missing in the

---

[*]   Equal contribution.
[†]   Corresponding Author.

39th Conference on Neural Information Processing Systems (NeurIPS 2025).

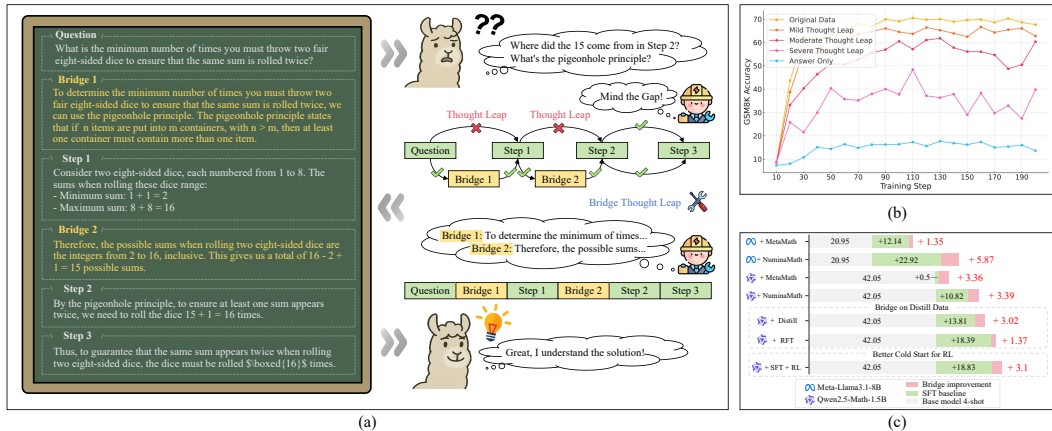

Figure 1: Overview of the Thought Leap phenomenon and our bridging approach. (a) Thought Leaps in CoT; (b) Negative impact on training; (c) Bridging leaps improves reasoning performance.

original CoT. Without these bridges, models struggle to follow how the pigeonhole principle applies to dice probability and where the value "15" comes from—gaps that require implicit knowledge that may be obvious to experts but create significant barriers for models during learning.

Thought Leaps arise naturally in CoT datasets sourced from educational materials and expert demonstrations. Human experts often omit steps they consider trivial based on their extensive background knowledge, unconsciously creating gaps that can impede effective learning. This phenomenon is particularly problematic for both human learners and LLMs that lack the implicit knowledge necessary to bridge these gaps.

Our preliminary investigations reveal the detrimental effects of Thought Leap on model performance. By systematically introducing varying degrees of step omissions in the MetaMathQA [12] dataset, from mild to severe, we demonstrate that Thought Leaps significantly undermine training effectiveness. As shown in Figure 1 (b), models trained on datasets with Thought Leaps exhibit substantially lower performance ceilings (up to 27.83% reduction in accuracy for severe leaps, see details in Appendix C) and slower convergence rates compared to those trained on complete reasoning chains. These findings align with recent research suggesting that structural disruptions in reasoning chains can be more harmful than factual inaccuracies [17, 18].

To address this challenge, we propose the **CoT Thought Leap Bridge Task**, which aims to automatically detect Thought Leaps and generate the necessary intermediate reasoning steps to restore coherence and completeness in CoT demonstrations. As illustrated in Figure 1 (a), our approach effectively transforms incomplete reasoning chains into coherent step-by-step solutions by inserting appropriate bridging content, allowing learners to follow the complete reasoning process. Our approach involves creating a specialized dataset (ScaleQM+) for the Thought Leap Bridge Task by systematically removing intermediate steps from the structured ScaleQuestMath [19] dataset and pairing incomplete reasoning chains with their complete counterparts. We then develop CoT-Bridge, a fine-tuned model based on Qwen2.5-Math-7B [20] specifically designed to identify and bridge Thought Leaps in mathematical reasoning. Finally, we apply CoT-Bridge to existing mathematical reasoning datasets to enhance their completeness and coherence, thereby improving the quality of training data for downstream models.

Our experimental results demonstrate that models fine-tuned on bridged datasets achieve significant performance improvements compared to those trained on the original datasets with Thought Leaps. As illustrated in Figure 1 (c), addressing the Thought Leap phenomenon leads to consistent improvements across different model architectures and datasets, with performance gains of up to +5.87% on NuminaMath and +3.36% on MetaMathQA. These results highlight the broad applicability of our method and its ability to enhance reasoning capabilities across various mathematical and logical reasoning benchmarks.

Furthermore, our approach functions as a plug-and-play enhancement module that can be integrated with other advanced techniques. As shown in Figure 1 (c), it can improve the quality of distilled

data (+3.02%) or provide better cold start models for Reinforcement Learning (+3.1%), thereby amplifying the effectiveness of existing methods for enhancing reasoning capabilities. In summary, our contributions are:

- To the best of our knowledge, we are the first to systematically identify and formalize the **Thought Leap** phenomenon in CoT reasoning. We introduce the **CoT Thought Leap Bridge Task** along with an evaluation framework for addressing this issue.

- We develop a specialized dataset (**ScaleQM+**) and a model (**CoT-Bridge**) for identifying and bridging Thought Leaps, demonstrating their effectiveness through comprehensive experiments.

- We apply CoT-Bridge to existing mathematical reasoning datasets, achieving significant performance improvements and demonstrating good generalization capabilities on out-of-domain logical reasoning benchmarks (+2.99%).

- We validate that our approach can function as a plug-and-play enhancement module, compatible with methods such as knowledge distillation and reinforcement learning, to further amplify model performance.

## 2 Method

In this section, we provide a detailed discussion of our approach. Section 2.1 presents a rigorous formalization of the Thought Leap phenomenon and CoT Thought Leap Bridge task. Section 2.2 describes the construction of the Thought Leap dataset and the training of the bridge model, along with a brief introduction to a variant method. Section 2.3 outlines the process of applying the bridge model to existing step-by-step datasets.

### 2.1 Task Formalization

Let $C^* = (Q, s_1^*, s_2^*, \ldots, s_m^*)$ represent an ideal complete CoT starting from question $Q$. In $C^*$, the transition from $Q$ to $s_1^*$, as well as transitions between any adjacent steps $(s_i^*, s_{i+1}^*)$, are both explicit and coherent. Defining a completeness function $V$, for $C^*$ we have $V(Q, s_1^*) =$ True and $\forall i \in [1, m-1], V(s_i^*, s_{i+1}^*) =$ True. This function captures whether the reasoning transition between adjacent steps is sufficiently detailed and logically sound.

Now consider a CoT $C = (s_0, s_1, s_2, \ldots, s_n)$ that successfully derives an answer $A$ (for convenience, we denote question $Q$ as $s_0$). If there exists at least one pair of adjacent steps $(s_k, s_{k+1})$ within $C$ that fails the completeness criterion, i.e., $V(s_k, s_{k+1}) =$ False, then a **Thought Leap** exists between $s_k$ and $s_{k+1}$. Such incompleteness indicates that necessary intermediate reasoning steps $S'_{\text{miss}} = (s'_{k.1}, s'_{k.2}, \ldots, s'_{k.j})$, where $j \geq 1$, have been omitted between $s_k$ and $s_{k+1}$. These omitted steps must satisfy three conditions: (1) completeness from $s_k$ to the first missing step: $V(s_k, s'_{k.1}) =$ True; (2) internal completeness within the missing sequence: $\forall i \in [1, j-1], V(s'_{k.i}, s'_{k.i+1}) =$ True; and (3) completeness from the last missing step to $s_{k+1}$: $V(s'_{k.j}, s_{k+1}) =$ True.

Based on this formalization, we define the **CoT Thought Leap Bridge Task** as a two-stage process: first identifying all adjacent step pairs $(s_k, s_{k+1})$ within $C$ that satisfy $V(s_k, s_{k+1}) =$ False, and then for each identified leap, generating the missing intermediate step sequence $S'_{\text{miss}}$ to bridge the gap as $s_k \oplus S'_{\text{miss}} \oplus s_{k+1}$. This task directly addresses the completeness aspect of reasoning, which is complementary to the more commonly studied factual accuracy dimension.

### 2.2 Thought Leap Bridging Dataset Construction and Bridge Model Training

To implement the CoT Thought Leap Bridge task, we created a specialized training dataset named **ScaleQM+** based on the structurally comprehensive ScaleQuestMath dataset. We chose ScaleQuest-Math as an approximation of ideal CoT because its reasoning structure is relatively complete and well-formed, making it suitable for systematic step removal. As illustrated in Figure 2, we start with complete reasoning chains $C^* = (s_0, s_1^*, \ldots, s_m^*)$ where $s_0 = Q$ is the question, and strategically remove intermediate steps to produce incomplete chains $C = (s_0, s_1, \ldots, s_n)$ containing Thought Leaps. The removed steps form the reference set of missing intermediate steps needed to bridge these leaps.

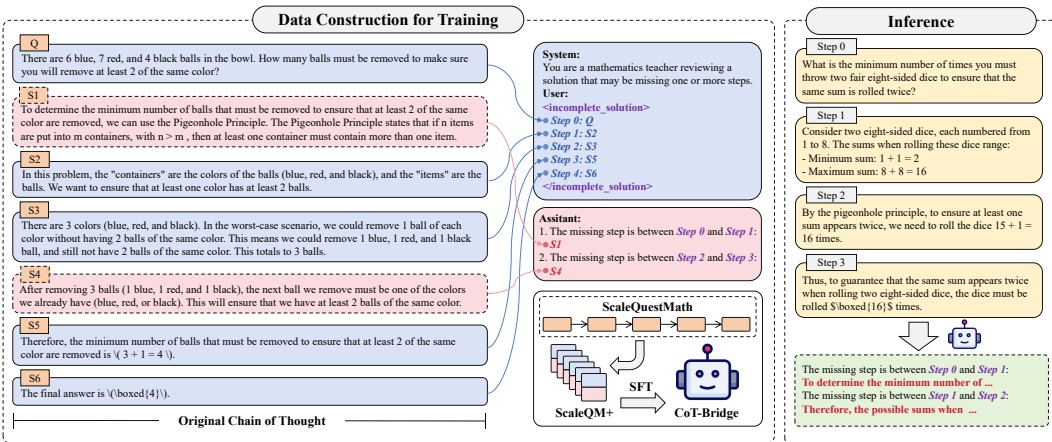

Figure 2: Illustration of our work. The left panel shows data construction for training, where we strategically remove intermediate steps (e.g., between Step 0 and Step 1, or Step 2 and Step 3) from complete reasoning chains in ScaleQuestMath to create ScaleQM+ with Thought Leaps. The right panel demonstrates inference, where CoT-Bridge identifies gaps and generates appropriate intermediate steps to restore coherence in reasoning.

Our step removal strategy follows several principles to create effective training examples. We always retain the final step $s_m^*$ to preserve the completeness of the answer, while allowing the initial reasoning step $s_1^*$ to be removed to help the model learn macro-planning skills. The number of steps to remove, $k_{del}$, scales with the length of the original chain: for shorter chains ($m \leq 10$), we remove 1-2 steps; for longer chains ($m > 10$), we remove 1-3 steps. Additionally, with a probability of 0.2, we retain the complete chain unchanged ($C = C^*$) to train the model to recognize when no bridging is needed.

For each modified chain $C$, we generate two essential training components: the ground-truth Thought Leap positions $\mathcal{L}_{gt}$ identifying where step deletion resulted in reasoning gaps, and the corresponding ground-truth missing steps $\mathcal{M}_{gt}$ consisting of the original removed steps. Our primary model, **CoT-Bridge**, learns the mapping $f : C \rightarrow (\hat{\mathcal{L}}, \hat{\mathcal{M}})$, taking an incomplete chain as input and producing both predicted leap positions and the corresponding missing steps. As a baseline for comparison, we also implement **CoT-Bridge-Random**, which learns $g : (C, \mathcal{L}_{gt}) \rightarrow \hat{\mathcal{M}}$, focusing solely on generating missing steps given the ground-truth leap positions.

In practice, we process the ScaleQuestMath dataset by segmenting reasoning chains using "\n\n" as delimiters and selecting examples with at least 6 steps ($m \geq 6$) to ensure sufficient complexity. This process yields 588k training samples with 10k examples held out for testing. We fine-tune CoT-Bridge from the Qwen2.5-Math-7B base model using standard instruction tuning techniques for one epoch. We provide details such as instruction templates and training parameters in Appendix E.

## 2.3 Data Augmentation with Bridge Model

After training, we apply CoT-Bridge to enhance existing mathematical reasoning datasets, specifically **MetaMathQA** and **NuminaMath-CoT**, creating improved versions named **MetaMath-Bridge** and **NuminaMath-Bridge** respectively. As shown in Figure 2, CoT-Bridge processes reasoning chains to identify gaps and generate appropriate bridging content.

Given an input reasoning chain $C = (s_0, s_1, \ldots, s_n)$, CoT-Bridge produces a set of predicted Thought Leap repairs $\{(k_i, \hat{S}'_{miss,k_i})\}_{i=1}^{N_{leap}}$, where each $k_i$ identifies a leap between steps $s_{k_i}$ and $s_{k_i+1}$, $N_{leap}$ represents the total number of leaps detected, and $\hat{S}'_{miss,k_i}$ is the sequence of generated intermediate steps. For each identified leap, we insert the generated steps between the corresponding original steps, performing $s_{k_i} \oplus \hat{S}'_{miss,k_i} \oplus s_{k_i+1}$ to create the bridged reasoning chain $C_{bridged}$.

To accommodate the format differences across datasets, we adapt our approach to the specific step delimiters used: **"\n"** for MetaMathQA and **"\n\n"** for NuminaMath-CoT. For the CoT-Bridge-Random variant, we provide randomly selected positions $k_{random}$ instead of model-identified leaps,

allowing us to evaluate the importance of accurate gap identification in improving reasoning quality. This approach enables us to assess both the leap detection and step generation aspects of the bridge task separately.

## 3 Experiments

### 3.1 Setup

To evaluate the generality and effectiveness of our proposed approach, we performed supervised fine-tuning (SFT) experiments using MetaMathQA, NuminaMath-CoT datasets and their bridged versions on representative base models. Specifically, we selected Meta-Llama3.1-8B [21] as a representative general-purpose model, and Qwen2.5-Math-1.5B [20] as a representative math-specialized model. To ensure comparability, we maintained a unified experimental configuration across all SFT experiments. The detailed training settings are provided in Appendix D.1.

### 3.2 Evaluation

We employed six benchmarks: GSM8K [15], MATH500 [22], and GaoKao2023EN [23] as basic-level benchmarks, and MathOdyssey [24], OlympiadBenchEN [25], and AMC23 [26] as advanced competition-level benchmarks. The evaluation was carried out using the vLLM [27] inference library. To ensure consistency, all models adopted identical generation parameters, employing greedy decoding with zero-shot prompting, without external tools. To mitigate evaluation variance, we sampled four outputs for each problem and computed average accuracy as the final metric. The detailed evaluation settings are provided in Appendix D.2.

### 3.3 Baselines

We establish a series of baselines for comparison. The most important baseline is standard SFT performed directly on the original datasets. In addition, we use a zero-shot bridging which method leveraging general-purpose LLMs (in this paper, Qwen2.5-Instruct-7B/72B [28]) to explore Thought Leap bridging without direct training. This baseline aims to assess the effectiveness of pure prompt engineering in bridging Thought Leaps, determining whether specialized detection and repair mechanisms are necessary. Furthermore, we use CoT-Bridge-Random to generated intermediate steps at randomly chosen positions, to evaluate the importance of accurate Thought Leap localization for effective bridging. As a reference, we consider the performance of base models in a 4-shot setting. Additionally, we present the results of SFT using GSM8K+MATH and MathInstruct.

### 3.4 Main Results

Table 1 presents our main experimental results across all benchmarks, base models, and training datasets. We observe several consistent patterns that highlight the effectiveness of our CoT Thought Leap Bridge approach.

**Bridging Thought Leaps consistently improves reasoning performance.** When comparing CoT-Bridge with Direct SFT, we observe substantial improvements across almost all configurations. The most significant gains are seen with Meta-Llama3.1-8B tuned on NuminaMath, where CoT-Bridge achieves an average improvement of +5.87%, with particularly impressive gains on competition-level benchmarks (+15.63% on AMC23). Similarly, for Qwen2.5-Math-1.5B on MetaMathQA, CoT-Bridge yields a +3.36% average improvement, with a remarkable +7% on MATH500. These results indicate that addressing reasoning completeness through bridging enhances model performance, especially on more challenging problems that require rigorous step-by-step reasoning.

**Accurate leap identification is crucial for effective bridging.** Comparing CoT-Bridge with CoT-Bridge-Random reveals the importance of precise gap detection. While CoT-Bridge consistently improves performance, CoT-Bridge-Random shows highly variable results, degrading performance on certain benchmarks. For instance, with Qwen2.5-Math-1.5B on NuminaMath, CoT-Bridge-Random decreases accuracy on GSM8K (-0.56%), GaoKao2023EN (-1.56%), and MathOdyssey (-3.68%), with only marginal average improvement (+0.64%). In contrast, CoT-Bridge achieves a substantial

| Dataset | Size | Method | Basic Level | | | Competition Level | | | Average |
|---|---|---|---|---|---|---|---|---|---|
| | | | GSM8K | MATH | GaoKao | Odyssey | Olympiad | AMC23 | |
| **Meta-Llama3.1-8B** | | | | | | | | | |
| / | / | 4-shot | 54.15 | 18.30 | 20.58 | 16.54 | 4.85 | 11.25 | 20.95 |
| GSM8K+MATH | 15k | Direct SFT | 65.09 | 19.25 | 21.69 | 18.48 | 5.07 | 12.50 | 23.68 |
| MathInstruct | 262k | Direct SFT | 68.16 | 23.60 | 25.52 | 25.06 | 5.89 | 7.50 | 25.96 |
| MetaMathQA | 395k | Direct SFT | 78.90 | 36.10 | 32.86 | 24.68 | 8.48 | 17.50 | 33.09 |
| | | QwenBridger-S | $81.10^{+2.20}$ | $34.85^{-1.25}$ | $30.52^{-2.34}$ | $22.67^{-2.01}$ | $9.26^{+0.78}$ | $7.50^{-10.00}$ | $30.98^{-2.11}$ |
| | | QwenBridger-L | $80.80^{+1.90}$ | $38.05^{+1.95}$ | $31.43^{-1.43}$ | $24.48^{-0.20}$ | $9.37^{+0.89}$ | $2.50^{-15.00}$ | $31.11^{-1.98}$ |
| | | CoT-Bridge-R | $80.46^{+1.56}$ | $38.05^{+1.95}$ | $\mathbf{33.57}^{+0.71}$ | $24.42^{-0.26}$ | $9.37^{+0.89}$ | $12.50^{-5.00}$ | $33.06^{-0.03}$ |
| | | CoT-Bridge | $\mathbf{81.14}^{+2.24}$ | $\mathbf{38.15}^{+2.05}$ | $33.12^{+0.26}$ | $\mathbf{25.97}^{+1.29}$ | $\mathbf{9.48}^{+1.00}$ | $\mathbf{18.75}^{+1.25}$ | $\mathbf{34.44}^{+1.35}$ |
| NuminaMath | 859k | Direct SFT | 84.86 | 51.45 | 49.03 | 36.56 | 21.30 | 20.00 | 43.87 |
| | | QwenBridger-S | $84.23^{-0.63}$ | $52.40^{+0.95}$ | $51.95^{+2.92}$ | $39.73^{+3.17}$ | $24.70^{+3.40}$ | $27.50^{+7.50}$ | $46.75^{+2.88}$ |
| | | QwenBridger-L | $85.25^{+0.39}$ | $54.20^{+2.75}$ | $51.62^{+2.59}$ | $39.08^{+2.52}$ | $25.33^{+4.03}$ | $35.00^{+15.00}$ | $48.41^{+4.54}$ |
| | | CoT-Bridge-R | $84.82^{-0.04}$ | $54.20^{+2.75}$ | $51.88^{+2.85}$ | $40.12^{+3.56}$ | $\mathbf{26.15}^{+4.85}$ | $33.75^{+13.75}$ | $48.50^{+4.63}$ |
| | | CoT-Bridge | $\mathbf{85.97}^{+1.11}$ | $\mathbf{56.80}^{+5.35}$ | $\mathbf{54.42}^{+5.39}$ | $\mathbf{40.76}^{+4.20}$ | $24.85^{+3.55}$ | $\mathbf{35.63}^{+15.63}$ | $\mathbf{49.74}^{+5.87}$ |
| **Qwen2.5-Math-1.5B** | | | | | | | | | |
| / | / | 4-shot | 79.00 | 48.05 | 45.52 | 38.18 | 19.07 | 22.50 | 42.05 |
| GSM8K+MATH | 15k | Direct SFT | 74.45 | 51.40 | 47.66 | 38.50 | 17.44 | 27.50 | 42.83 |
| MathInstruct | 262k | Direct SFT | 70.96 | 48.90 | 46.49 | 40.89 | 16.22 | 20.00 | 40.58 |
| MetaMathQA | 395k | Direct SFT | 81.01 | 49.60 | 46.62 | 38.63 | 18.19 | 21.25 | 42.55 |
| | | QwenBridger-S | $\mathbf{81.58}^{+0.57}$ | $51.30^{+1.70}$ | $46.69^{+0.07}$ | $38.63^{-0.00}$ | $18.52^{+0.33}$ | $25.63^{+4.38}$ | $44.04^{+1.49}$ |
| | | QwenBridger-L | $81.01^{-0.00}$ | $53.63^{+4.03}$ | $49.22^{+2.60}$ | $38.70^{+0.07}$ | $18.85^{+0.66}$ | $27.50^{+6.25}$ | $44.82^{+2.27}$ |
| | | CoT-Bridge-R | $\mathbf{81.58}^{+0.57}$ | $53.65^{+4.05}$ | $48.12^{+1.50}$ | $39.02^{+0.39}$ | $18.22^{+0.03}$ | $25.63^{+4.38}$ | $44.37^{+1.82}$ |
| | | CoT-Bridge | $81.39^{+0.38}$ | $\mathbf{56.60}^{+7.00}$ | $\mathbf{49.61}^{+2.99}$ | $\mathbf{39.66}^{+1.03}$ | $\mathbf{19.44}^{+1.25}$ | $\mathbf{28.75}^{+7.50}$ | $\mathbf{45.91}^{+3.36}$ |
| NuminaMath | 859k | Direct SFT | 83.62 | 63.90 | 57.40 | 46.77 | 33.04 | 32.5 | 52.87 |
| | | QwenBridger-S | $83.23^{-0.39}$ | $64.20^{+0.30}$ | $58.25^{+0.85}$ | $46.96^{+0.19}$ | $32.04^{-1.00}$ | $35.00^{+2.50}$ | $53.28^{+0.41}$ |
| | | QwenBridger-L | $82.81^{-0.81}$ | $66.25^{+2.35}$ | $57.79^{+0.39}$ | $46.64^{-0.13}$ | $32.70^{-0.34}$ | $40.00^{+7.50}$ | $54.37^{+1.50}$ |
| | | CoT-Bridge-R | $83.06^{-0.56}$ | $65.20^{+1.30}$ | $55.84^{-1.56}$ | $43.09^{-3.68}$ | $33.89^{+0.85}$ | $40.00^{+7.50}$ | $53.51^{+0.64}$ |
| | | CoT-Bridge | $\mathbf{84.61}^{+0.99}$ | $\mathbf{68.05}^{+4.15}$ | $\mathbf{59.29}^{+1.89}$ | $\mathbf{47.16}^{+0.39}$ | $\mathbf{34.11}^{+1.07}$ | $\mathbf{45.00}^{+12.50}$ | $\mathbf{56.26}^{+3.39}$ |

Table 1: Main results (%) on mathematical benchmarks. MATH, GaoKao, Odyssey, and Olympiad correspond to the MATH500, GaoKao2023EN, MathOdyssey, and OlympiadBenchEN benchmarks, respectively. QwenBridger-S and QwenBridger-L represent zero-shot bridging based on Qwen2.5-Instruct-7B and Qwen2.5-Instruct-72B, respectively. CoT-Bridge-R stands for CoT-Bridge-Random.

+3.39% average gain under the same setting. This stark difference highlights that merely inserting additional steps without strategic placement can disrupt reasoning coherence, whereas targeted bridging at identified leaps enhances logical flow and ultimately improves reasoning quality.

**Zero-shot bridging shows promise but lacks consistency.** Although zero-shot bridging demonstrates certain effectiveness on specific tasks, its overall performance still falls short compared to CoT-Bridge. Taking the LLaMA + NuminaMath setup as an example, the 72B bridge model improves the average accuracy by 4.54% over standard SFT, whereas CoT-Bridge further boosts it to 5.87%. Notably, the noise introduced by zero-shot bridge can degrade model performance in some scenarios. Under the LLaMA + MetaMathQA configuration, the 72B model shows decreased performance on GaoKao2023EN, MathOdyssey, and AMC23 by 1.43%, 0.20%, and 15.00%, respectively, leading to an overall average drop of 1.98%. This effect is more pronounced when using the weaker 7B model. In contrast, CoT-Bridge achieves more robust improvements under the same configuration, raising the average accuracy by 1.35%, highlighting its superior quality control and structural adaptability.

## 4 Analysis

### 4.1 Plug-and-Play Integration

CoT-Bridge can serve as a plug-and-play enhancement module that seamlessly integrates into existing training pipelines while delivering consistent performance improvements. To evaluate its adaptability and benefits across different training paradigms, we applied it to two representative scenarios: (1) improving the quality of generated data in knowledge distillation and rejection sampling, and (2)

| Dataset | Method | GSM8K | MATH | GaoKao | Odyssey | Olympiad | AMC23 | Average |
|---------|--------|-------|------|--------|---------|----------|-------|---------|
| Distill | Direct SFT | 81.86 | 68.15 | 60.84 | 48.13 | 33.00 | 39.37 | 55.23 |
|         | CoT-Bridge | $82.52^{+0.66}$ | $71.50^{+3.35}$ | $66.43^{+5.59}$ | $49.16^{+1.03}$ | $34.89^{+1.89}$ | $45.00^{+5.63}$ | $58.25^{+3.02}$ |
| Reject Sampling | Direct SFT | 83.36 | 74.90 | 64.94 | 51.81 | 37.63 | 50.00 | 60.44 |
|                 | CoT-Bridge | $83.74^{+0.38}$ | $75.25^{+0.35}$ | $67.47^{+2.53}$ | $51.87^{+0.06}$ | $39.41^{+1.78}$ | $53.13^{+3.13}$ | $61.81^{+1.37}$ |

Table 2: Model performance after bridging distilled and rejection-sampled data.

assessing whether models fine-tuned on CoT-Bridge-enhanced data perform better in subsequent RL stages compared to models trained on original data.

We used Qwen2.5-Instruct-72B to generate data via distillation and rejection sampling on GSM8K and MATH training sets. For rejection sampling, we sampled 4 responses per prompt (temperature = 1.0, top-p = 1.0) and retained the correct answers, and then bridged them using CoT-Bridge. As shown in Table 2, models fine-tuned on the bridged data achieve superior performance: average accuracy improved from 55.23% (distill) / 60.44% (rejection sampling) to 58.25% / 61.81%, respectively.

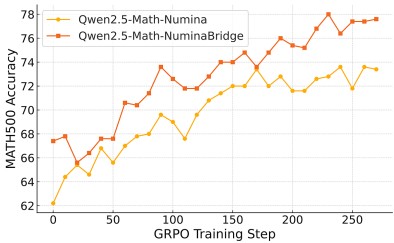

Figure 3: Model accuracy over training steps on MATH500.

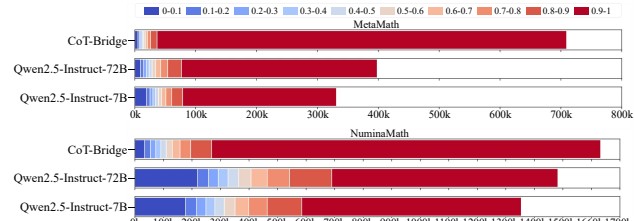

Figure 4: PRM scores of Qwen2.5-Instruct-7B/72B on CoT-Bridge for MetaMathQA and NuminaMath.

In the reinforcement learning setup, we continued training the Qwen2.5-Math-1.5B models that were fine-tuned on NuminaMath and NuminaMath-Bridge in the main experiments, using the GRPO [29] algorithm and the DAPO-Math-17K [30] dataset. Additional training details are provided in the Appendix G. Figure 3 shows the training curves of models, demonstrating that our method provides both a higher starting point and a better final performance. As shown in Table 9 from Appendix G.2, the model trained with NuminaMath-Bridge achieved an RL accuracy of 63.98%, outperforming that trained with NuminaMath at 60.88%. This result also surpasses the officially released Qwen2.5-Math-Instruct-1.5B [20] model, which was also trained with GRPO algorithm.

## 4.2 Evaluation of Thought Leap Bridge Task

To evaluate the capability of various methods in CoT Thought Leap Bridge Task, we constructed a standardized evaluation framework on ScaleQM+ test set, covering leap identification and generation quality. Calculation methods for these metrics are detailed in Appendix H. Case study is detailed in Appendix I. As shown in Table 3, CoT-Bridge significantly outperforms other methods in localization precision (78.02%), redundancy rate (1.61%) and overall metric (76.15%). Qwen2.5-Instruct-72B notably outperforms the 7B model, indicating that

| Method | Similarity | Position | | | Overall↑ |
|--------|------------|----------|------|------|----------|
|        |            | Pre↑ | Rec↑ | Red↓ | |
| Qwen2.5-Instruct-7B | / | 14.15 | 12.04 | 34.13 | 10.54 |
| Qwen2.5-Instruct-72B | / | 33.99 | 33.64 | 33.73 | 31.12 |
| CoT-Bridge | / | **78.02** | 78.37 | **1.61** | **76.15** |
| Full-position | 1 | 20.96 | **79.64** | 79.04 | 75.72 |
|  | 0.95 | 23.42 | 75.07 | 76.57 | 71.22 |
|  | 0.90 | 24.81 | 59.21 | 74.37 | 55.9 |
|  | 0.85 | 19.34 | 30.17 | 63.87 | 28.26 |
|  | 0.80 | 7.47 | 8.65 | 32.51 | 7.84 |

Table 3: Performance of different bridging methods on ScaleQM+ test set. **Pre** stands for Precision, **Rec** stands for Recall and **Red** stands for Redundancy.

stronger generative models possess certain zero-shot localization and restoration capabilities. We also evaluated an exhaustive approach based on **full-position generation** and **similarity filtering**. This method uses CoT-Bridge-Random to generate candidate content between all adjacent steps, filtering them using similarity thresholds. While this approach demonstrates high recall capability,

it exhibits an extremely high redundancy rate. The substantial redundant content may cause LLMs to learn low-quality repetitive patterns during fine-tuning, thereby compromising reasoning ability. Additionally, this strategy is highly sensitive to similarity thresholds, incurs significant computational costs, and is difficult to scale in practice.

## 4.3 Evaluation of Out-of-Domain Reasoning Capability

To verify the generalization capability of our method in Out-of-Domain(OOD) reasoning, we conducted evaluations on five out-of-domain logical reasoning datasets (FOLIO [31], LogicQA [32], ProofWriter [33], ReClor [34], RuleTaker [35]) that were not used in training. These datasets cover formal logic, factual deduction, and textual entailment reasoning types. In the evaluation, we used XFinder [36] to automatically extract answers from model responses, with outputs that failed to yield valid answers (such as those deviating from the prompt or exhibiting logical loops) counted as errors and separately tallied.

| Bridge Method | Mertic | FOLIO | LogicQA | PW | ReClor | RuleTaker | Average |
|---|---|---|---|---|---|---|---|
| **NuminaMath+Meta-Llama3.1-8B** | | | | | | | |
| No Bridge | Accuracy ↑ | 68.15 | 34.33 | 59.09 | 47 | 54.5 | 52.61 |
| | Invalid ↓ | 1.48 | 6.3 | 0.51 | 2 | 0.31 | 2.12 |
| GapBridge | Accuracy ↑ | **74.07**$^{+5.92}$ | **35.64**$^{+1.31}$ | **61.52**$^{+2.43}$ | **50.20**$^{+3.20}$ | **56.57**$^{+2.07}$ | **55.60**$^{+2.99}$ |
| | Invalid ↓ | 0.74 | 4.92 | 0.2 | 2.2 | 0.1 | 1.63 |
| **NuminaMath+Qwen2.5-Math-1.5B** | | | | | | | |
| No Bridge | Accuracy ↑ | **74.07** | 29.72 | 55.84 | 37.6 | 53.05 | 50.06 |
| | Invalid ↓ | 1.48 | 4.53 | 0.3 | 1.2 | 0.72 | 1.65 |
| GapBridge | Accuracy ↑ | 71.11$^{-2.96}$ | **33.10**$^{+3.38}$ | **58.38**$^{+2.54}$ | **39.00**$^{+1.40}$ | **53.67**$^{+0.62}$ | **51.05**$^{+0.99}$ |
| | Invalid ↓ | 1.48 | 3.46 | 0.3 | 1.4 | 0 | 1.33 |

Table 4: Performance of NuminaMath and its bridged version on logical reasoning benchmarks.

As shown in Table 4, we compared the performance of models trained on NuminaMath and NuminaMath-Bridge. Results indicate that CoT-Bridge improved OOD reasoning performance (Meta-Llama3.1-8B: ↑2.99%, Qwen2.5-Math-1.5B: ↑0.99%) while reducing invalid response rates. We attribute the performance improvement to the refined intermediate steps introduced by CoT-Bridge, which help models master general reasoning structures and enhance generalization capabilities.

## 4.4 Analysis of Bridging Positions

Leap bridging can enhance reasoning capabilities of LLMs, but whether complementary content bridged at different structural positions contributes substantially still lacks systematic analysis. Therefore, we categorized the bridging content in CoT-Bridge by its position in the reasoning chain as *begin* (reasoning starting point, planning strategy), *middle* (key calculations and logical progression), and *end* (result verification and summary).

Statistics show that *middle* type bridgings dominate (MetaMathQA: 66.34%; NuminaMath: 55.52%), followed by *begin*, with *end* accounting for a relatively small proportion (approximately 2%). To evaluate the specific role of each type of content, we designed position component ablation experiments: removing one type of bridging (*begin* / *middle* / *end*) from the CoT-Bridge data while keeping the rest unchanged, and fine-tuning Qwen2.5-Math-1.5B under identical training settings. As shown in Table 5, experimental results indicate that removal leads to performance degradation. This suggests that all three types of completion have a positive effect on model performance. We believe that *begin* bridgings help clarify problem objectives and establish solution frameworks; *middle* bridgings support critical links in reasoning paths; and *end* bridgings, although proportionally smaller, help strengthen logical closure and result reasonability in certain tasks.

## 4.5 Process Supervision Scoring and Noise Impact Analysis

To measure the bridging quality of various methods, we introduced the process supervision model Qwen2.5-Math-PRM-7B to score the intermediate steps (range 0–1) and analyzed the score distribu-

| Delete Pos | Num | Ratio | GSM8K | MATH | GaoKao | Odyssey | Olympiad | AMC23 | Average |
|---|---|---|---|---|---|---|---|---|---|
| **Qwen2.5-Math-1.5B+MetaMath** | | | | | | | | | |
| / | / | / | 81.39 | **56.60** | 49.61 | 39.66 | **19.44** | **28.75** | 45.91 |
| - begin | 225873 | 31.84 | **82.71**$^{+1.32}$ | 55.85$^{-0.75}$ | 49.55$^{-0.06}$ | **41.09**$^{+1.43}$ | 19.22$^{-0.22}$ | 22.50$^{-6.25}$ | 45.15$^{-0.76}$ |
| - middle | 470541 | 66.34 | 80.97$^{-0.42}$ | 52.45$^{-4.15}$ | 49.35$^{-0.26}$ | 37.14$^{-2.52}$ | 17.26$^{-2.18}$ | 27.50$^{-1.25}$ | 44.11$^{-1.80}$ |
| - end | 12908 | 1.82 | 81.14$^{-0.25}$ | 55.75$^{-0.85}$ | **49.94**$^{+0.33}$ | 37.02$^{-2.64}$ | 18.78$^{-0.66}$ | 28.12$^{-0.63}$ | 45.13$^{-0.78}$ |
| **Qwen2.5-Math-1.5B+NuminaMath** | | | | | | | | | |
| / | / | / | **84.61** | **68.05** | **59.29** | **47.16** | **33.44** | **45.00** | **56.26** |
| - begin | 694887 | 42.57 | 82.60$^{-2.01}$ | 64.85$^{-3.20}$ | 58.67$^{-0.62}$ | 45.48$^{-1.68}$ | 32.96$^{-0.48}$ | 38.75$^{-6.25}$ | 53.89$^{-2.37}$ |
| - middle | 906168 | 55.52 | 83.51$^{-1.10}$ | 63.55$^{-4.50}$ | 56.43$^{-2.86}$ | 47.16$^{-0.00}$ | 28.30$^{-5.14}$ | 38.13$^{-6.87}$ | 52.85$^{-3.41}$ |
| - end | 31115 | 1.91 | 82.79$^{-1.82}$ | 64.05$^{-4.00}$ | 55.97$^{-3.32}$ | 46.51$^{-0.65}$ | 27.48$^{-5.96}$ | 42.50$^{-2.50}$ | 53.22$^{-3.04}$ |

Table 5: Performance variation after removing each bridging component.

| Threshold | GSM8K | MATH | GaoKao | Odyssey | Olympiad | AMC23 | Average |
|---|---|---|---|---|---|---|---|
| **MetaMath+Qwen2.5-Math-1.5B** | | | | | | | |
| / | **81.39** | 56.60 | 49.61 | **39.66** | 19.44 | 28.75 | 45.91 |
| prm $< 0.1$ | 81.22$^{-0.17}$ | **57.00**$^{+0.40}$ | 50.45$^{+0.84}$ | 37.40$^{-2.26}$ | **19.70**$^{+0.26}$ | 29.38$^{+0.63}$ | 45.86$^{-0.05}$ |
| prm $< 0.3$ | 80.99$^{-0.40}$ | 56.90$^{+0.30}$ | **51.88**$^{+2.27}$ | 37.79$^{-1.87}$ | 18.33$^{-1.11}$ | **32.50**$^{+3.75}$ | **46.40**$^{+0.49}$ |
| prm $< 0.5$ | 80.53$^{-0.86}$ | 54.75$^{-1.85}$ | 49.09$^{-0.52}$ | 37.86$^{-1.80}$ | 18.78$^{-0.66}$ | 25.00$^{-3.75}$ | 44.34$^{-1.57}$ |
| **NuminaMath+Qwen2.5-Math-1.5B** | | | | | | | |
| / | **84.61** | **68.05** | 59.29 | **47.16** | 33.44 | **45.00** | **56.26** |
| prm $< 0.1$ | 83.28$^{-1.33}$ | 65.50$^{-2.55}$ | 59.09$^{-0.20}$ | 46.19$^{-0.97}$ | **34.33**$^{+0.89}$ | 41.88$^{-3.12}$ | 55.04$^{-1.22}$ |
| prm $< 0.3$ | 83.85$^{-0.76}$ | 63.00$^{-5.05}$ | **59.35**$^{+0.06}$ | 46.38$^{-0.78}$ | 33.07$^{-0.37}$ | 45.00$^{-0.00}$ | 55.11$^{-1.15}$ |
| prm $< 0.5$ | 83.26$^{-1.35}$ | 65.95$^{-2.10}$ | 58.05$^{-1.24}$ | 46.06$^{-1.10}$ | 33.56$^{+0.12}$ | 41.88$^{-3.12}$ | 54.79$^{-1.47}$ |

Table 6: Performance after removing low-scoring steps (evaluated by Qwen2.5-Math-PRM-7B).

tion. Table 11 in Appendix K and Figure 4 present the PRM score distribution of the bridged steps. Results show that steps generated by CoT-Bridge have higher quality compared to other methods. On the NuminaMath dataset, CoT-Bridge's proportion in the high-quality (range 0.9–1) reaches 83.51%, significantly higher than 72B Fill (53.45%) and 7B Fill (56.8%). Furthermore, CoT-Bridge's proportion in the low-quality (range 0–0.1) is only 2.08%, compared to 14.81% and 13.12% for 72B and 7B Fill, respectively. We also used DeepSeek-R1 [6] to score the CoT comprehensively, CoT bridged by CoT-Bridge scoring higher than the original chains. The scoring prompt, results and examples are provided in the Appendix J.

To further verify whether low-quality steps generated by CoT-Bridge affect model training effectiveness, we set different thresholds based on PRM scores to denoise the bridged data and conducted SFT training on Qwen2.5-Math-1.5B accordingly. As shown in Table 6, results indicate that removing low-scoring steps has limited impact on model performance, suggesting that noise introduced by bridging has minimal effect on training. On the NuminaMath dataset, after removing steps with PRM scores below 0.1, accuracy decreased from 56.26% to 55.04%; while on MetaMathQA, removing steps with scores below 0.3 slightly improved performance from 45.91% to 46.40%. We attribute this phenomenon to two factors: first, the proportion of noise introduced by CoT-Bridge itself is already very low; second, the PRM model may misjudge in complex reasoning scenarios, with certain steps marked as low-scoring still possessing heuristic and training value.

## 5 Conclusion

In this paper, we addressed the critical issue of Thought Leaps in CoT reasoning by introducing the CoT Thought Leap Bridge Task and developing the CoT-Bridge model trained on ScaleQM+ dataset. Our approach automatically detects and fills these reasoning gaps, demonstrably enhancing the completeness and coherence of reasoning chains. Comprehensive experiments revealed that models tuned on these bridged datasets lead to significant performance improvements and better generalization to out-of-domain reasoning tasks. Furthermore, CoT-Bridge acts as an effective plug-

and-play module, improving outcomes in knowledge distillation and reinforcement learning, thereby highlighting the substantial benefits of rectifying thought leaps for more robust and capable large language models.

## Acknowledgement

This work is supported by the National Natural Science Foundation of China (No. 62436007), the Key Research and Development Program of Zhejiang Province, China (No. 2024C01034), the Fundamental Research Funds for the Central Universities (226-2024-00170), MOE Engineering Research Center of Digital Library, CCF-Baidu Open Fund and ZJU Kunpeng&Ascend Center of Excellence.

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

# A    Related Works

## A.1    Methods for Enhancing Mathematical Reasoning

Researchers have explored various reasoning enhancement methods based on CoT, broadly categorized into inference-time methods and training-based methods.

Inference-time approaches improve reasoning by adopting advanced decoding strategies or utilizing extra computational resources. Self-Consistency (SC) [37] enhances robustness by sampling multiple reasoning paths and applying majority voting. Search-based methods, including Tree-of-Thought (ToT) [38] and Monte Carlo Tree Search (MCTS) [39], systematically explore reasoning spaces for optimal solutions. Recent test-time expansion techniques, such as multi-round thinking [40] and s1 budget forcing [41], dynamically increase computation to boost performance. In addition, inference-time intervention methods [42, 43] enhance reasoning performance by manipulating hidden states.

Training-based approaches internalize enhanced reasoning abilities through parameter updates during model training. A common strategy involves fine-tuning models on datasets with explicitly detailed intermediate reasoning steps. Additionally, training paradigms incorporating self-reflection [44], self-verification, self-breaking [45] and self-correction have gained attention, teaching models to recognize and rectify errors autonomously. Frameworks like StepCo [46], S2R [47], and ReVISE [48] integrate these metacognitive capabilities through supervised, reinforcement, or preference learning. In particular, recent advances highlight that reinforcement learning plays an especially crucial role [49–52] in enhancing the overall capabilities of large models. We leverage CoT-Bridge to enhance the structural completeness of existing datasets, enabling models to acquire finer-grained reasoning patterns during training and to be seamlessly integrated into existing training pipelines.

## A.2    High-Quality Mathematical Datasets for Training

Developing high-quality datasets is essential for enhancing mathematical reasoning capabilities in LLMs. MetaMathQA [12] consists of 395k problem-solution pairs derived by reformulating and reverse-engineering existing datasets like GSM8K [15] and MATH [16]. MathInstruct [13] aggregates 13 existing math datasets and employs GPT-4 [53] to synthesize 260k examples containing both Chain-of-Thought and Program-of-Thought solutions. NuminaMath-CoT [14] compiles data from examinations, competitions, and Q&A communities, resulting in a dataset of 860k problem-solution pairs. ScaleQuestMath [19] introduces a zero-shot data generation framework for mathematical questions using small open-source models, resulting in a scalable dataset of about 1 million samples. DAPO [30] developed DAPO-Math-17K, comprising 17k standardized integer-formatted math problems sourced primarily from platforms such as AoPS for effective reward model evaluation.

Beyond creating new datasets, extensive research has focused on data augmentation (e.g., MathFusion [54], PersonaMathQA [55], MathFimer [56]) and selective strategies (e.g., QaDS [57], DELIFT [58]) to enhance data efficiency and training effectiveness. CoT-Bridge contributes by bridging Thought Leap issues in existing mathematical reasoning datasets, thus improving their overall data quality.

# B    Limitations

**Potential Noise in Reasoning Steps:**    CoT-Bridge can't guarantee that all bridged reasoning steps are entirely correct, which may inevitably introduce a certain level of noise during automated data construction. Although our analysis in Section 4.5 suggests that this noise has a limited impact on overall model performance, its presence remains a non-negligible concern.

**Lack of Validation on Larger-Scale Models:**    Due to computational resource constraints, our experiments focus primarily on small to medium-sized models, and we have not yet conducted evaluations on larger models such as 32B or 72B-scale LLMs. This may limit the verification of our method's generalization capability on very large-scale pretrained models. Future work could explore its performance and adaptability on larger architectures.

**Training Limited to Mathematical Data:**    CoT-Bridge is trained exclusively on ScaleQuestMath in mathematical domain. While we demonstrate promising OOD generalization to logical reasoning tasks that are relatively close to mathematics, the method remains largely domain-specific. It is worth noting, however, that CoT-Bridge is theoretically generalizable and can potentially be applied to other multi-step reasoning tasks in domains such as law, medicine, and scientific QA. Future research could further investigate its adaptability across diverse task types.

# C    Preliminary Experiment

## C.1    Settings

To simulate varying degrees of Thought Leap, we created modified versions of the MetaMathQA dataset by removing intermediate reasoning steps. Considering the structure of MetaMathQA, we use **"\n"** to segment individual steps within CoT. The specific configurations are as follows:

- The final step which contains the answer explanation is always retained.
- **Mild, Moderate, Severe Thought Leap:** We randomly remove 1, 2, or 3 intermediate steps, respectively, if sufficient steps are available.
- **Extreme Thought Leap:** All intermediate reasoning steps are removed and only the final answer explanation is preserved.

We adopt the same training configuration as described in Section 3.1 using Meta-Llama3.1-8B. Each model is trained for 200 steps, with evaluation performed on the GSM8K every 10 steps. For convenience, answer verification is conducted using Math-Verify only.

## C.2    Results

We present the detailed results from Figure 1(b) in Table 7.

| Training Step | Original Data | Mild Thought Leap | Moderate Thought Leap | Severe Thought Leap | Answer Only |
|---|---|---|---|---|---|
| 10 | 8.64 | 8.79 | 8.64 | 8.57 | 7.35 |
| 20 | 43.67 | 38.71 | 33.2 | 25.78 | 8.04 |
| 30 | 62.32 | 56.03 | 40.33 | 21.53 | 10.77 |
| 40 | 60.96 | 55.34 | 46.4 | 29.95 | 15.09 |
| 50 | 62.77 | 55.42 | 50.87 | 40.33 | 14.4 |
| 60 | 64.9 | 60.88 | 50.57 | 35.71 | 16.37 |
| 70 | 68.08 | 63.61 | 52.69 | 35.18 | 14.86 |
| 80 | 66.94 | 64.67 | 55.57 | 37.98 | 16.22 |
| 90 | 69.9 | 66.03 | 56.94 | 40.03 | 16.22 |
| 100 | 69.06 | 64.52 | 60.5 | 37.76 | 16.38 |
| 110 | 70.51 | 63.68 | 57.16 | 48.37 | 17.29 |
| 120 | 69.74 | 66.41 | 61.03 | 37.15 | 15.62 |
| 130 | 69.98 | 65.28 | 61.79 | 36.39 | 17.66 |
| 140 | 68.92 | 63.91 | 57.77 | 37.83 | 16.83 |
| 150 | 69.6 | 62.47 | 56.1 | 29.04 | 16.22 |
| 160 | 69.83 | 66.64 | 56.03 | 38.29 | 17.36 |
| 170 | 68.61 | 64.29 | 54.66 | 29.72 | 15.01 |
| 180 | 70.28 | 65.5 | 48.75 | 32.9 | 15.39 |
| 190 | 68.61 | 66.03 | 50.49 | 27.45 | 16 |
| 200 | 67.63 | 62.77 | 60.35 | 39.8 | 13.65 |

Table 7: Accuracy of Meta-Llama3.1-8B on GSM8K over training steps using data with varying degrees of Thought Leaps.

# D   Experimental Details

## D.1   Model Training Settings

We utilized LlamaFactory [59] as SFT training framework. The initial learning rate was set to $1 \times 10^{-5}$ with a warm-up ratio of 0.1, and cosine scheduling was used to gradually reduce the learning rate to zero. The maximum sequence length was set to 8192 tokens, with a global batch size of 128. We trained models for 3 epochs on MetaMathQA [12] and for 2 epochs on NuminaMath-CoT [14], as preliminary experiments indicated no further improvement from additional epochs. All SFT experiments were conducted using 8 Ascend H910B-64G.

For SFT training template, we use the default style template in LlamaFactory:

> **Prompt Template**
>
> **System:** You are a math problem solver. You should think step by step.
> **Human:** <Question>
> **Assistant:** <Answer>

## D.2   Evaluation Settings

The maximum token limit for generation was set to 2048, and prompt templates remained consistent with training ones. It is noteworthy that, although we set the decoding temperature parameter to zero, vLLM outputs still exhibited some randomness. For answer extraction and comparison, we employed the Math-Verify [60] tool. Given limitations of math-verify in handling complex expressions, responses failing initial verification were subsequently validated using DeepSeek-R1 [6]. All model evaluations were performed using 4 NVIDIA A100-40G GPUs.

Referring to OpenR1 [61], we use the following prompt template:

> **Prompt Template**
>
> You are a mathematical answer validator. You will be provided with a mathematical problem and you need to compare the answer in the reference solution, and the final answer in a model's solution to determine if they are equivalent, even if formatted differently.
>
> PROBLEM:
> <problem>
>
> REFERENCE SOLUTION:
> <answer>
>
> MODEL'S SOLUTION:
> <generation>
>
> Focus ONLY on comparing the final mathematical answer provided by the model while ignoring differences in:
>
> - Formatting (e.g., \boxed{} vs plain text)
> - Multiple choice formatting (e.g., "A" vs full solution)
> - Order of coordinate pairs or solutions
> - Equivalent mathematical expressions or notation variations
> - If the model's answer is nonsense, return "Verdict: AMBIGUOUS"
>
> Start with a brief explanation of your comparison (2-3 sentences). Then output your final answer in one of the following formats:
>
> - "Verdict: EQUIVALENT"
> - "Verdict: DIFFERENT"
> - "Verdict: AMBIGUOUS"

# E  ScaleQM+ Construction and CoT-Bridge training

## E.1  ScaleQM+ Template

> **Prompt Template**
>
> **System:**
> You are a mathematics teacher reviewing a solution that may be missing one or more steps.
> Your task is to:
> 1. Identify all points in the logical flow where a step is missing. For each missing step, specify exactly between which two consecutive steps it should be placed.
> 2. Provide the complete missing step(s) with necessary explanations and equations. The solution may be missing multiple steps or might be complete. The steps in the solution are labeled from Step 0 (problem statement) to Step N.
> For each missing step, please format your response as follows::
> *Missing Step X:*
> *The missing step should be placed between Step Y and Step Y+1.*
> *The missing step is:*
> *[Write the complete missing step here with necessary explanations and equations]*
>
> If there are no missing steps, please output:
> *No missing steps.*
> **Human:**
> <incomplete_solution>
> Step 0:
> <question statement>
> Step 1:
> ...
> Step N:
> ...
> </incomplete_solution>
> **Assistant:**
> <Answer>

## E.2  Variant Template

> **Prompt Template**
>
> **System:**
> You are a mathematics teacher reviewing a solution that appears to be missing one step. Given the position of the missing step, your task is to fill in the missing step.
> The steps in the solution are labeled from Step 0 (problem statement) to Step N.
> Please format your response as:
> *The missing step is:*
> *[Write the complete missing step here with necessary explanations and equations]*
> **Human:**
> **There is a missing step between Step X and Step X+1.**
> <incomplete_solution>
> Step 0:
> <question statement>
> Step 1:
> ...
> Step N:
> ...
> </incomplete_solution>
> **Assistant:**
> <Answer>

## E.3  Discussion of Hyperparameter Settings

We set the minimum CoT length $m$ to 6, based on the following considerations. CoTs with too few steps should be avoided, as identifying Thought Leaps in such cases becomes trivially easy. For example, in a two-step CoT,

the final step (the answer) cannot be removed, so only the first step is eligible for deletion. This makes it very easy for the model to identify the location of the Thought Leap, which lies directly between the question and the answer.

Regarding the number of steps to delete, we avoid removing too many steps. For instance, in a 6-step CoT, deleting 4 steps would leave too little context, making the task resemble generating a new solution based on the final answer rather than bridging a partial reasoning chain.

As for the probability of retaining the original CoT without modification, this parameter is introduced to avoid introducing redundancy into already complete reasoning chains.

### E.4 CoT-Bridge Training Settings

We broadly follow the setup in D.1, but train for only 1 epoch with a global batch size of 1024, considering the large amount of data and the primary goal of learning the structural integrity of CoT rather than introducing new knowledge.

## F Evaluation Benchmarks

### F.1 Mathematics Benchmarks

- **GSM8K**: This benchmark consists of 1,319 grade-school math word problems, primarily used to evaluate a model's ability to perform multi-step arithmetic reasoning.

- **MATH500**: A curated subset of 500 representative problems from the larger MATH dataset, designed to reflect the original dataset's coverage across diverse mathematical topics and difficulty levels.

- **Gaokao2023EN**: This benchmark includes 385 English-translated questions from the 2023 Chinese Gaokao (college entrance exam) mathematics section. It tests a model's ability to solve complex problems at the advanced high school level.

- **MathOdyssey**: A benchmark of 387 problems spanning difficulty levels from high school to early undergraduate mathematics. It emphasizes evaluating a model's performance on problems requiring deeper understanding and more complex reasoning steps.

- **OlympiadBenchEN**: A collection of 675 problems at the difficulty level of international mathematical olympiads. Known for their non-standard formats and high demands on creative problem-solving, these problems serve as a rigorous test of advanced mathematical reasoning capabilities.

- **AMC23**: This benchmark contains 40 problems selected from the 2023 American Mathematics Competitions (AMC), offering a challenging testbed for evaluating models in a standardized competition setting.

### F.2 Logical Reasoning Benchmarks

- **FOLIO**: This benchmark consists of 1,430 expert-crafted natural language inference instances, each annotated with first-order logic (FOL) expressions. It is designed to assess a model's ability to translate between natural language and formal logic, as well as perform multi-step reasoning over complex logical structures.

- **LogicQA**: Collected from the Chinese National Civil Servant Examination, this benchmark targets logical reasoning in reading comprehension tasks.

- **ProofWriter**: Composed of multiple small rulebases, each containing natural language descriptions of facts and rules, along with questions labeled as true, false, or unknown. This benchmark supports evaluating a model's ability to generate reasoning chains and perform multi-step logical inference.

- **ReClor**: Derived from standardized graduate-level entrance exams such as the LSAT and GMAT, ReClor evaluates a model's performance on complex logical reasoning questions.

- **RuleTaker**: Built from synthetically generated rules and facts, this benchmark tests whether a model can simulate deductive reasoning. It evaluates the model's ability to draw logical conclusions from natural language rules.

# G  RL Training

## G.1  Settings

All RL experiments were conducted using the veRL [62] training framework. We employed Math-Verify for answer verification. We set the initial learning rate to $1 \times 10^{-6}$ and used a global batch size of 512. The maximum response length was limited to 4096 tokens. During rollout, 4 samples were generated per input. KL regularization was disabled, and evaluation was performed with zero temperature every 10 epochs on MATH500. We use the same template in Appendix D.1.

## G.2  Results

We present the detailed results from Figure 3(b) in Table 8. For simplicity, only Math-Verify is used for answer verification without further applying DeepSeek-R1.

| Training Step | Qwen2.5-Math-Numina | Qwen2.5-Math-NuminaBridge |
|---|---|---|
| 0 | 62.2 | 67.4 |
| 10 | 64.4 | 67.8 |
| 20 | 65.4 | 65.6 |
| 30 | 64.6 | 66.4 |
| 40 | 66.8 | 67.6 |
| 50 | 65.6 | 67.6 |
| 60 | 67 | 70.6 |
| 70 | 67.8 | 70.4 |
| 80 | 68 | 71.4 |
| 90 | 69.6 | 73.6 |
| 100 | 69 | 72.6 |
| 110 | 67.6 | 71.8 |
| 120 | 69.6 | 71.8 |
| 130 | 70.8 | 72.8 |
| 140 | 71.4 | 74 |
| 150 | 72 | 74 |
| 160 | 72 | 74.8 |
| 170 | 73.4 | 73.6 |
| 180 | 72 | 74.8 |
| 190 | 72.8 | 76 |
| 200 | 71.6 | 75.4 |
| 210 | 71.6 | 75.2 |
| 220 | 72.6 | 76.8 |
| 230 | 72.8 | 78 |
| 240 | 73.6 | 76.4 |
| 250 | 71.8 | 77.4 |
| 260 | 73.6 | 77.4 |
| 270 | 73.4 | 77.6 |

Table 8: Model accuracy on MATH500 over GRPO training steps.

| Model | Method | GSM8K | MATH | GaoKao | Odyssey | Olympiad | AMC23 | Average |
|---|---|---|---|---|---|---|---|---|
| Qwen2.5-Math-Instruct-1.5B | / | **84.80** | 75.80 | 65.50 | **54.52** | 38.10 | **60.00** | 63.12 |
| Oat-Zero-1.5B | Dr. GRPO | 83.62 | 74.20 | **69.61** | 52.71 | 37.60 | 53.00 | 61.79 |
| Qwen2.5-Math-1.5B | GRPO | 82.71 | 74.60 | 64.94 | 49.10 | 35.85 | 50.00 | 59.33 |
| Qwen2.5-Math-Numina | GRPO | 84.31 | 74.80 | 62.34 | 51.94 | 39.41 | 52.50 | 60.88 |
| Qwen2.5-Math-NuminaBridge | GRPO | 84.08$^{-0.23}$ | **78.20**$^{+3.40}$ | 67.01$^{+4.67}$ | 54.26$^{+2.32}$ | **40.30**$^{+0.89}$ | **60.00**$^{+7.50}$ | **63.98**$^{+3.10}$ |

Table 9: Reinforcement learning results. Qwen2.5-Math-Numina and Qwen2.5-Math-NuminaBridge refer to the Qwen2.5-Math-1.5B models fine-tuned on NuminaMath and NuminaMath-Bridge respectively in the main experiments. We also present the results of Qwen2.5-Math-1.5B trained directly on DAPO-Math-17K using GRPO, along with the results of Qwen2.5-Math-Instruct-1.5B[20] and Oat-Zero-1.5B [63] for reference.

# H    Metrics for CoT Thought Leap Bridge Task

## H.1    Leap Position Identification Metrics

Let the ground-truth set of leap positions be $G = \{g_1, \ldots, g_{|G|}\}$, and the predicted set be $P = \{p_1, \ldots, p_{|P|}\}$. If a predicted position $p_j$ exactly matches a ground-truth position $g_i$, it is counted as a True Positive (TP). The metrics are defined as:

$$\text{Precision} = \frac{\text{TP}}{|P|}, \quad \text{Recall} = \frac{\text{TP}}{|G|}$$

To measure the proportion of redundant predictions, we define the redundancy rate as:

$$\text{Redundancy} = \frac{|P| - \text{TP}}{|P|}$$

## H.2    Overall Metric

To evaluate both leap positions locating ability and the quality of the generated step, we introduce the **Position-aware BERTScore**. This metric only compares text at successfully matched positions. Let the reference text at a matched position be $t_i$ and the generated text be $\hat{t}_i$, the Position-aware BERTScore is defined as:

$$\text{BERTScore}_{\text{pos-aware}} = \frac{1}{|G|} \sum_{i=1}^{\text{TP}} \text{BERTScore}(\hat{t}_i, t_i)$$

Here, BERTScore is computed using the RoBERTa-large model and measures the semantic similarity between each pair of texts $(\hat{t}_i, t_i)$.

# I    Case Study: CoT Thought Leap Bridge

Following the example in Figure 1(a), we compared the zero-shot bridging results of Qwen2.5-Instruct-7B/72B, GPT-4o, and Gemini-2.5-PRO.

---

**CoT-Bridge**

Missing Step 1:
The missing step should be placed between Step 0 and Step 1.
The missing step is:
To determine the minimum number of times you must throw two fair eight-sided dice to ensure that the same sum is rolled twice, we can use the pigeonhole principle. The pigeonhole principle states that if n items are put into m containers, with n > m, then at least one container must contain more than one item.

Missing Step 2:
The missing step should be placed between Step 1 and Step 2.
The missing step is:
Therefore, the possible sums when rolling two eight-sided dice are the integers from 2 to 16, inclusive. This gives us a total of 16 - 2 + 1 = 15 possible sums.

---

**Qwen2.5-7B-Instruct**

Missing Step 1:
The missing step should be placed between Step 0 and Step 1.
The missing step is:
The possible sums when rolling two eight-sided dice range from 2 to 16, which gives us 15 different possible sums.

---

### Qwen2.5-72B-Instruct

Missing Step 1:
The missing step should be placed between Step 1 and Step 2.
The missing step is:
The possible sums when rolling two eight-sided dice are 2, 3, 4, ..., 16. There are 15 possible sums in total.

Missing Step 2:
The missing step should be placed between Step 2 and Step 3.
The missing step is:
According to the pigeonhole principle, if we roll the dice 16 times, there are 15 possible sums, so at least one of these sums must appear at least twice.

### Gemini-2.5-PRO

Missing Step 1:
The missing step should be placed between Step 1 and Step 2.
The missing step is:

Now, we need to determine the total number of distinct sums possible. The sums are all the integers from 2 to 16, inclusive. These distinct sums will be our "pigeonholes."
To count the number of possible sums:
Number of distinct sums=(Maximum Sum-Minimum Sum)+1
Number of distinct sums=(16-2)+1=14+1=15

So, there are 15 unique possible sums (the pigeonholes) when rolling two eight-sided dice.

### GPT-4o

Missing Step 1:
The missing step should be placed between Step 1 and Step 2.
The missing step is:
We need to count how many distinct sums can result from rolling two eight-sided dice. The possible sums range from 2 to 16, which gives:
$$
16 - 2 + 1 = 15 \text{ distinct possible sums.}
$$

As can be seen in the case study:

- Although a planning-style bridge between the question and step 1 was missing, GPT-4o, Gemini-2.5-PRO, and Qwen2.5-72B-Instruct all identified that "15" lacked a calculation explanation and bridged the gap in the appropriate position.
- Qwen2.5-72B-Instruct generated a redundant step (almost identical to step 2). This phenomenon is quite common and aligns with its 33.73% redundancy rate shown in Table 3.
- The consistency of the content and position bridged by Qwen2.5-7B-Instruct is not good.

# J CoT Score

## J.1 Score Template

Since scoring a single response with DeepSeek-R1 can result in high variance, we instead compare the original CoT and its bridged version for the same question side-by-side.

---

**DeepSeek-R1 Score Template**

You are a math reasoning expert. Please evaluate the coherence of the given two Chain of Thoughts (CoTs) based on the provided question.
Perform an overall assessment of the CoTs based on the following five dimensions:
1. Problem Understanding: Does the CoT accurately identify the core of the question and interpret its requirements appropriately?
2. Planning: Does it show evidence of having a structured plan or certain degree of foresight in approaching the problem?
3. Logical Coherence: Is the reasoning step-by-step, consistent, and free of major logical leaps?
4. Detail Elaboration: Does it include detailed intermediate calculations, formula derivations, or other necessary expansions?
5. Conclusion Support: Does it provide a clear summary or verification to support the final conclusion?

- Question
<problem>
- CoT 1
<answer1>
- CoT 2
<answer2>

[Scoring Criteria Description]
- 10 points: Outstanding across all dimensions. Demonstrates deep understanding, clear planning, rigorous step-by-step reasoning, detailed intermediate calculations or derivations, and strong conclusion verification. Rarely achieved; reflects expert-level CoT.
- 8 points: Strong performance with only minor flaws. Reasoning is coherent and mostly complete, with evident planning and elaboration. May miss slight details or final verification, but overall very solid.
- 6 points: Adequate but uneven. Shows basic understanding and partial planning. Some steps are under-explained or loosely connected; may lack full elaboration or support for conclusion.
- 4 points: Weak overall. Reasoning is fragmented or inconsistent, with limited detail and minimal evidence of structured planning. Key steps are missing or poorly justified.
- 2 points: Very poor. Little to no logical structure or elaboration; response lacks meaningful connection to the question.
- 0 points: No relevant reasoning. Off-topic or fundamentally flawed throughout.
Note: Scores may also be assigned in odd numbers (e.g., 5, 7, 9) to reflect responses that fall between two defined levels. For instance, a score of 7 indicates performance stronger than the 6-point criteria but not fully reaching the 8-point standard.

[Output Format]
- Justification:
Logical Coherence Analysis: <Your brief analysis>
- Score:
CoT 1 overall score: <Integer between 0-10>
CoT 2 overall score: <Integer between 0-10>

A brief analysis is sufficient.

---

## J.2 Score Example



**Example**

Okay, let's tackle this problem. The user wants me to evaluate two Chain of Thoughts (CoTs) for the question about the minimum number of throws needed to ensure a repeated sum with two eight-sided dice. The scoring is based on five dimensions: Problem Understanding, Planning, Logical Coherence, Detail Elaboration, and Conclusion Support.

First, I need to understand the question thoroughly. The key here is the application of the pigeonhole principle. The possible sums when rolling two 8-sided dice range from 2 (1+1) to 16 (8+8). That gives 15 possible sums. According to the pigeonhole principle, to guarantee a repeat, you need one more than the number of possible sums, so 15 + 1 = 16 throws. Both CoTs arrive at this answer, but I need to assess how well each explains the reasoning.

Looking at CoT 1: It starts by identifying the minimum and maximum sums correctly. Then it directly applies the pigeonhole principle, stating 15 +1 =16. The conclusion is there, but the explanation is very brief. It doesn't mention how many possible sums there are, just states the range. The step from the range to 15 possible sums is implicit. A reader might wonder why 15 instead of 16-2=14. So, CoT 1 skips explaining the count of possible sums, which is a crucial step. The logical steps are there but not fully elaborated.

CoT 2 is more detailed. It explicitly states the pigeonhole principle, defines the range of sums, calculates the number of possible sums as 15 (16-2+1), then adds 1. This makes the reasoning clearer. The explanation of why there are 15 possible sums is important because it's a common point of confusion (inclusive counting). CoT 2 addresses that, making it more thorough.

Now, evaluating each dimension for both CoTs:

For Problem Understanding: Both correctly identify the core of the question, which is applying the pigeonhole principle. CoT 2 does a better job by explicitly stating the principle and the count of sums.

Planning: CoT 2 structures the answer step-by-step, starting with the principle, then calculating possible sums, then applying the formula. CoT 1 jumps to the answer without explaining the count of sums, which is a gap in planning.

Logical Coherence: Both are logically correct, but CoT 1 has a minor leap in assuming the number of possible sums. CoT 2 connects each step more clearly.

Detail Elaboration: CoT 2 elaborates on the number of sums, which CoT 1 omits. This makes CoT 2's explanation more complete.

Conclusion Support: Both reach the correct conclusion, but CoT 2's conclusion is better supported by the preceding steps.

Scoring: CoT 1 might get a 6 or 7 because it's correct but lacks some details. CoT 2 is more thorough, so probably an 8 or 9. Since CoT 2 explains each step clearly without missing key points, it's closer to the 8-9 range. CoT 1 is adequate but not as detailed, so maybe a 7. However, according to the criteria, 8 is for strong performance with minor lapses. CoT 2 doesn't have lapses, so maybe 9? Wait, the scoring criteria say 10 is outstanding. CoT 2 is thorough but maybe not "outstanding" because it's a straightforward application. Alternatively, since it's complete, maybe 10. But the example given in the criteria for 10 is "precise, step-by-step reasoning" with "careful attention to clarity". CoT 2 does that. CoT 1 is missing the explanation of the number of sums, so maybe a 6 or 7. Let me check again.

CoT 1: Problem Understanding: Correct. Planning: Minimal, skips the count of sums. Logical Coherence: Correct but with a leap. Detail Elaboration: Missing the count explanation. Conclusion: Correct. So maybe 6 or 7. The criteria for 6 say "intermediate steps present but not fully explained". CoT 1 has the steps but skips the count. So 6. CoT 2: All steps explained, so 10? But the example for 10 says "smooth and complete". CoT 2 is complete. Alternatively, maybe 9 if there's a minor point missing. But I don't see any. It explains the pigeonhole principle, calculates the sums correctly, adds 1. So 10. But the user's example answer might have given 8 and 10. Wait, the user's example in the criteria for 10 is "outstanding across all dimensions". CoT 2 is thorough, but perhaps not outstanding. Hmm. Alternatively, maybe 9. But given that it's correct and explains all steps, 10. CoT 1 gets 6 because it's adequate but missing the count explanation. </think>

- Justification:

Logical Coherence Analysis: CoT 1 correctly applies the pigeonhole principle but skips explaining the count of possible sums (15), leaving a minor gap. CoT 2 explicitly calculates the number of sums (16 - 2 + 1 = 15) and links each step clearly, ensuring full logical continuity.

- Score:

CoT 1 overall score: 7

CoT 2 overall score: 10

**Note:** CoT 1 earns a 7 for adequate reasoning but lacks explicit calculation of the sum count. CoT 2 achieves a 10 by thoroughly addressing all dimensions, including precise step-by-step elaboration and seamless logical flow.



The question, CoT1, and CoT2 here are the same as those shown in the example in Figure 1(a).

## J.3   Score Result

We randomly sampled 1,000 CoT pairs from both MetaMath and NuminaMath for scoring, and the results are shown in Table 10.

|  | Original CoT | Bridged CoT |
|---|---|---|
| Metamath | 7.85 | 8.44 |
| Numinamath | 7.49 | 8.22 |

Table 10: DeepSeek-R1 scoring results for the original CoT and the bridged CoT.

## K   PRM Score

We present the data from Figure 4 in Table 11.

| Method | Metric | 0-0.1 | 0.1-0.2 | 0.2-0.3 | 0.3-0.4 | 0.4-0.5 | 0.5-0.6 | 0.6-0.7 | 0.7-0.8 | 0.8-0.9 | 0.9-1 |
|---|---|---|---|---|---|---|---|---|---|---|---|
| **MetaMathQA** | | | | | | | | | | | |
| GapBridge | num | 5021 | 2516 | 2149 | 2228 | 2401 | 2765 | 3560 | 5045 | 10812 | 672825 |
|  | percent | 0.71 | 0.35 | 0.3 | 0.31 | 0.34 | 0.39 | 0.5 | 0.71 | 1.52 | 94.85 |
| Qwen2.5-Instruct-72B | num | 9528 | 4946 | 4410 | 4516 | 4891 | 5990 | 8172 | 11505 | 22577 | 321582 |
|  | percent | 2.39 | 1.24 | 1.11 | 1.13 | 1.23 | 1.5 | 2.05 | 2.89 | 5.67 | 80.78 |
| Qwen2.5-Instruct-7B | num | 19349 | 5557 | 4706 | 4372 | 4714 | 5287 | 7023 | 9568 | 18096 | 252466 |
|  | percent | 5.84 | 1.68 | 1.42 | 1.32 | 1.42 | 1.6 | 2.12 | 2.89 | 5.46 | 76.24 |
| **NuminaMath** | | | | | | | | | | | |
| GapBridge | num | 33923 | 20643 | 18397 | 17857 | 18783 | 21589 | 27041 | 37375 | 73499 | 1363063 |
|  | percent | 2.08 | 1.26 | 1.13 | 1.09 | 1.15 | 1.32 | 1.66 | 2.29 | 4.5 | 83.51 |
| Qwen2.5-Instruct-72B | num | 219586 | 38135 | 34371 | 34360 | 36910 | 43698 | 56699 | 78320 | 147988 | 792281 |
|  | percent | 14.81 | 2.57 | 2.32 | 2.32 | 2.49 | 2.95 | 3.82 | 5.28 | 9.98 | 53.45 |
| Qwen2.5-Instruct-7B | num | 177687 | 38694 | 32492 | 31388 | 32955 | 37837 | 47592 | 65059 | 121298 | 769034 |
|  | percent | 13.12 | 2.86 | 2.4 | 2.32 | 2.43 | 2.79 | 3.51 | 4.8 | 8.96 | 56.8 |

Table 11: The count and proportion of bridge steps falling into different PRM score intervals for Qwen2.5-Instruct-7B/72B and CoT-Bridge on MetaMath and NuminaMath.

