# OpenReview forum: "Mind the Gap: Bridging Thought Leap for Improved Chain-of-Thought Tuning"
_NeurIPS.cc/2025/Conference — NeurIPS 2025 poster_

### Official Review · Reviewer_Ayux · 2025-06-24

**Clarity:** 3
**Significance:** 3
**Originality:** 3
**Rating:** 4
**Confidence:** 4

**Summary:**

This paper proposes to augment existing SFT in LLMs by thought bridge data: i.e., CoT data that's specifically constructed to be missing intermediate reasoning steps, to enhance the model's ability in reasoning.

The authors did comprehensive experiments and show that the constructed ScaleQM+ data can help improve model's performance compared to regular SFT.

**Questions:**

- To clarify, for the results in Table 1, how exactly is the ScaleQM+ dataset used? Is it used to create the CoT-Bridge model, then the model was used to augment MetaMathQA and NuminaMath, and then SFT on those augmented datasets were done?
In that case, is CoT-bridge utilizing one additional dataset ScaleQuestMath? Does the baseline (direct SFT) use MetaMathQA / NuminaMath + ScaleQuestMath, or just the former only?

**Ethical Concerns:**

["NO or VERY MINOR ethics concerns only"]

**Final Justification:**

The experiments ablating ScaleQM+ confirm that the bridging can indeed boost model performance, in addition to just dataset gains. Overall I'm leaning positive for this paper and would like to keep my current positive rating.

**Limitations:**

Not discussed. Please see weaknesses above

**Paper Formatting Concerns:**

Minor:
- line  111 and 136, figure number missing

**Quality:**

3

**Strengths And Weaknesses:**

Strengths:
- The paper is overall well-written.
- The overall idea about model should learn to bridge thought gaps is quite interesting, and the authors managed to construct a dataset to quantify this behaviour, and improve the model's reasoning performance significantly.
- Experiments are performed over two models, LLama and Qwen, and over several math datasets, to show the effectiveness of CoT-Bridge.

Weaknesses:
- For the results in Table 1, do CoT-Bridge use an additional dataset ScaleQuestMath where the baselines don't?
- This paper is well-motivated and to support its claims, I would expect seeing data with real thought leaps being bridged in training. The current way of processing the data is kind of artificial, where you know where the missing step is and the model is asked to predict that exact step. In practice, I wonder given any dataset (possibly with thought leaps but you don't know where that is), how can you enable the model to bridge such thought leaps?

---

> ### Author Rebuttal · Authors · 2025-07-31
>
> **Dear Reviewer Ayux:**
>
> Thank you for reviewing our paper and providing valuable feedback. Below, we present our responses to the weaknesses you have mentioned:
>
> > Question:
> To clarify, for the results in Table 1, how exactly is the ScaleQM+ dataset used? Is it used to create the CoT-Bridge model, then the model was used to augment MetaMathQA and NuminaMath, and then SFT on those augmented datasets were done? In that case, is CoT-bridge utilizing one additional dataset ScaleQuestMath? Does the baseline (direct SFT) use MetaMathQA / NuminaMath + ScaleQuestMath, or just the former only?
>
> **Response to Question:**
>
> Thank you for your question. We would like to clarify the CoT-Bridge and baseline setup.
>
> 1. **CoT-Bridge Setup:**
>    - We first train the CoT-Bridge model on ScaleQM+ (derived from a subset of ScaleQuestMath, ~588k samples from the original 1M, as mentioned in Section 2.2, lines 128-132)
>    - Then apply CoT-Bridge to augment MetaMathQA/NuminaMath
>    - Finally perform SFT on the augmented datasets
>
> 2. **Baseline Setup:**
>    The baseline model (Direct SFT) is trained only on MetaMathQA or NuminaMath.
>
> We understand your concern regarding the fairness of this comparison. **We will address this issue through additional experiments in our response to Weakness 1.**
>
> > Weakness 1:
> For the results in Table 1, do CoT-Bridge use an additional dataset ScaleQuestMath where the baselines don't?
>
> **Response to Weakness 1:**
>
> Thank you for your insightful observation regarding the fairness of comparison.
>
> - Indeed, CoT-Bridge utilizes ScaleQM+ (derived from ScaleQuestMath) during the augmentation process of MetaMathQA and NuminaMath. However, this utilization differs from direct SFT using ScaleQuestMath. CoT-Bridge is designed to learn **structural patterns** for identifying and bridging thought gaps rather than directly acquiring mathematical knowledge from the dataset.
>
> - Simply incorporating the 588k ScaleQuestMath subset (used to construct ScaleQM+, as mentioned in Section 2.2, lines 128-132) into the baseline's SFT training would **also create an imbalance** by significantly increasing the baseline's training samples.
>
> To ensure equitable comparison, we conducted additional experiments where both the baseline and CoT-Bridge-augmented setup incorporated this 588k subset. Due to the limited time and computational resources available during the rebuttal period, we could not conduct comprehensive experiments across all model sizes and datasets as in our main study. We focused our testing on Meta-Llama3.1-8B + NuminaMath as a representative configuration. As shown in the table below, the performance improvement is **still significant and consistent**.
>
> Results on Meta-Llama3.1-8B + NuminaMath:
>
> | Method | Size | GSM8K | MATH | GaoKao | Odyssey | Olympiad | AMC23 | Average |
> |-|-|-|-|-|-|-|-|-|
> | NuminaMath (from Table 1) | 859k | 84.86 | 51.45 | 49.03 | 36.56 | 21.30 | 20.00 | 43.87 |
> | NuminaMath-Bridged (from Table 1) | 859k | 85.97 | 56.80 | 54.42 | 40.76 | 24.85 | 35.63 | 49.74 |
> | NuminaMath + 588k subset | 859k + 588k | 86.12 | 55.60 | 54.29 | 36.43 | 22.37 | 32.50 | 47.89 |
> | NuminaMath-Bridged + 588k subset | 859k + 588k | 87.19 | 59.20 | 56.88 | 41.60 | 25.48 | 37.50 | 51.31 |
> | Improvement | - | +1.07 | +3.60 | +2.59 | +5.17 | +3.11 | +5.00 | +3.42 |
>
> We sincerely appreciate your observation. We plan to include these additional results in our paper to enhance its rigor.
>
> > Weakness 2:
> This paper is well-motivated and to support its claims, I would expect seeing data with real thought leaps being bridged in training. The current way of processing the data is kind of artificial, where you know where the missing step is and the model is asked to predict that exact step. In practice, I wonder given any dataset (possibly with thought leaps but you don't know where that is), how can you enable the model to bridge such thought leaps?
>
> **Response to Weakness 2:**
>
> Thank you for your kind words of our motivation. We would like to clarify how our approach works in practice. CoT-Bridge does **not require any external guidance about where thought leaps occur** during inference. When processing any reasoning chain (e.g., from MetaMathQA or NuminaMath) that potentially contains thought leaps, CoT-Bridge simultaneously:
> 1) Identifies **WHERE** thought leaps occur (position detection)
> 2) Generates **WHAT** content to bridge the gaps (content generation)
>
> - As formally defined in Section 2.1 (lines 123-127), CoT-Bridge learns the mapping $f: C → (L̂, M̂)$, taking a reasoning chain $C$ as input and autonomously producing both the leap positions $L̂$ and corresponding missing steps $M̂$.
>
> - Actually, Figure 1(a) in the paper demonstrate this capability with a real-world example. In Figure 1(a), CoT-Bridge independently identifies the thought leaps (between Question - Step 1 and between Step 1 - Step 2) and generates appropriate bridging content (Bridge 1 and Bridge 2) without external guidance.
>
> - The prompt used in inference is consistent with the one in Section E.1, which shows that CoT-Bridge does not require users to provide the bridge positions.
>
> We hope this clarification addresses your concern regarding the practical application of CoT-Bridge to datasets with unknown thought leap positions. If you have any further questions, we would be happy to answer them.
>
> Sincerely,
> CoT-Bridge Authors

---

> > ### Comment · Reviewer_Ayux · 2025-08-04
> >
> > Thanks for the additional experiments. The experiments ablating ScaleQM+ confirm that the bridging can indeed boost model performance, in addition to just dataset gains. Please add those results in the final version. Overall I'm leaning positive for this paper and would like to keep my current positive rating.

---

> > > ### Author Response · Authors · 2025-08-06
> > >
> > > Thank you for the feedback. We're glad the new ablation results have addressed your concern and confirmed the bridging effect. We’ll include them in the final version.We appreciate your leaning positive on the paper.

---

### Official Review · Reviewer_Wqes · 2025-06-29

**Clarity:** 3
**Significance:** 2
**Originality:** 3
**Rating:** 4
**Confidence:** 3

**Summary:**

To address the problem of thought leap in language model's Chain-of-Thought (CoT) reasoning where model skips intermediate steps, this paper proposes to finetune a thought-bridge model on a curated thought leap bridging dataset in order to augment existing mathematical reasoning datasets with sufficient intermediate steps. The authors experiment on six math reasoning datasets models and demonstrate that models tuned on the bridged datasets improves performance.

**Questions:**

1. When you augment the dataset by filling the missing intermediate skips, you are also increasing the number of tokens used for training which also improves the model performance due to the scaling law with respect to the number of tokens. Is it a fair comparison to compare augmented datasets with more tokens to original datasets?

2. While you create a specialized training dataset ScaleQM+ and remove intermediate steps to produce incomplete chains, are you assuming that ScaleQM+ is then free of thought leap issue? If so, why not directly train the models on this dataset? In other words, the (human) efforts you used to construct the thought-leap-free dataset in order to train the bridging model can be otherwise used to complete existing math reasoning datasets such as MATH500, then why would we prefer to do the former?

**Ethical Concerns:**

["NO or VERY MINOR ethics concerns only"]

**Final Justification:**

The issues of model scalable, impact on RL , and information-theoretic efficiency have been resolved.

**Limitations:**

Yes

**Quality:**

2

**Strengths And Weaknesses:**

## Strength:

### Identification of novel task

This works identifies the task of CoT Thought Leap Bridge that presents an automatic approach to detect reasoning leaps and generate missing steps. This task is useful to enhance the dataset construction of LLM training and has rich applicability to various training pipelines such as knowledge distillation and RLHF.

## Weakness:

### Model scalability concerns

In order to demonstrate improvement on Meta-Llama3.1-8B and Qwen2.5-Math-1.5B, this work uses 10x larger language models  (72B) for data augmentation. This raises a model scalability concern whether the advertised improvement is scalable to larger model sizes. If improvements are limited to smaller models, the contribution is less significant, especially given the abundance of existing methods that generate synthetic data using larger language models.

### Limited effects on RL and narrow evaluation

Since CoT-Bridge operates on augmenting thought processes within offline data, its benefits are mostly realized during supervised fine-tuning (SFT), rather than in reinforcement learning (particularly outcome-based RL), which is more prevalent and impactful in finetuning reasoning models. Although the work shows that CoT-Bridge can provide a better SFT initialization for RL (Appendix G), the evaluation is limited: it is demonstrated on only one dataset and with a single small model, restricting the generalizability and strength of the claims.

### Information-theoretical inefficiency

The performance gains from SFT-trained models stem from the bridge model’s ability to provide thought-bridging supervision, which in turn relies on training with a large, customized dataset (ScaleQM+, derived from the structurally rich ScaleQuestMath dataset). However, this process is information-theoretically inefficient, due to the data processing inequality. There are at least two alternative and potentially more efficient strategies: (1) redirecting human annotation efforts towards filling in missing reasoning steps directly within the training datasets, or (2) training models directly on the comprehensive customized dataset, rather than relying on an intermediate bridge model.

---

> ### Author Rebuttal · Authors · 2025-07-31
>
> **Dear Reviewer Wqes:**
>
> Thank you for the time and effort you have dedicated to reviewing our paper. Below, we present our responses to the weaknesses and questions you have mentioned:
>
> > Weakness 1: Model scalability concerns
>
> **Response to Weakness 1:**
>
> Thank you for pointing out the potential model scalability concern. We would like to respectfully clarify that our core data augmentation tool **CoT-Bridge is a 7B-scale model** fine-tuned from Qwen2.5-Math-7B as described in Section 2.2 (lines 130-132), rather than "10x larger language models (72B)". **Actually, the 72B models are used to highlight the efficiency and practicality of CoT-Bridge-7B.**
>
> - **Zero-Shot Baseline:** In Tables 1 and 3, we use the 72B model QwenBridger-L (Qwen2.5-Instruct-72B without fine-tuning) for zero-shot bridging as a strong baseline. The experiment results clearly demonstrate that our CoT-Bridge-7B model significantly outperforms 10x larger model both in terms of enhanced dataset quality and bridging task performance.
>
> - **Enhancing Data from Larger Models:** In Section 4.1, Table 2, we use the 72B model to generate distilled data. The purpose of this experiment is to demonstrate that even data generated by 10x larger models still exists thought leaps. Our 7B CoT-Bridge model can effectively enhance the high-quality but still imperfect data, leading to further performance improvements (+3.02%).
>
> Furthermore, we strongly agree that the bridge model size is indeed worth exploring. We believe that the thought gap bridge task requires **a task-specialized model** rather than necessarily a larger-size one with strong capabilities. Since the bridge task has access to the original solution CoT, it doesn't require an especially large model compared to directly solving the problem. To test this hypothesis, we fine-tuned Qwen2.5-Math-1.5B on ScaleQM+ to create CoT-Bridge-1.5B. We then used this smaller model to enhance NuminaMath and performed SFT on Meta-Llama3.1-8B with the enhanced dataset. As shown in the table below, the performance using CoT-Bridge-1.5B for bridging is not significantly different from our main results in the paper.
>
> Results on Meta-Llama3.1-8B + NuminaMath:
>
> | Method | GSM8K | MATH | GaoKao | Odyssey | Olympiad | AMC23 | Average |
> |-|-|-|-|-|-|-|-|
> | Direct SFT | 84.86 | 51.45 | 49.03 | 36.56 | 21.30 | 20.00 | 43.87 |
> | CoT-Bridge-7B | 85.97 | 56.80 | 54.42 | 40.76 | 24.85 | 35.63 | 49.74 |
> | CoT-Bridge-1.5B | 85.67 | 55.60 | 54.03 | 40.05 | 25.19 | 35.00 | 49.26 |
>
> > Weakness 2.1: Limited effects on RL
>
> **Response to Weakness 2.1**
>
> Thank you for raising this important point about the relationship between our method and RL. We would like to clarify that the main purpose of our RL analysis experiments in Section 4.1 was to verify whether better SFT initialization can lead to **a higher performance ceiling** for subsequent RL training.
>
> The results in Table 9 of our paper (Appendix G.2) demonstrate that models initialized with CoT-Bridge-enhanced data achieved 63.98% accuracy after RL, compared to 60.88% for models without bridging (+3.1% improvement). Furthermore, when compared to directly applying RL to Qwen2.5-Math-1.5B (59.33%), our approach shows a more substantial +4.65% improvement.
>
> These findings align with DeepSeek-R1 [1], which demonstrates that models with SFT initialization (DeepSeek-R1) achieve superior reasoning performance compared to directly trained with RL from scratch (DeepSeek-R1-Zero).
>
> [1] DeepSeek-R1: Incentivizing Reasoning Capability in LLMs via Reinforcement Learning
>
> > Weakness 2.2: Narrow evaluation
>
> **Response to Weakness 2.2**
>
> We understand your concern about the comprehensiveness of our experiments. Due to computational resource constraints and the limited time available during the rebuttal period, we conducted additional RL experiments using another smaller dataset (MATH training set, difficulty levels 3-5) with the Llama3.1-8B fine-tuned on NuminaMath and NuminaMath-Bridged. Additionally, we included results from SimpleRL-Zoo [2] on direct RL training of Llama3.1-8B for your reference.
>
> As shown in the table below, the results continue to show that using datasets enhanced with thought bridges for RL leads to a higher performance ceiling across all benchmarks.
>
> Results:
>
> | Model | GSM8K | MATH | GaoKao | Odyssey | Olympiad | AMC23 | Average |
> |-|-|-|-|-|-|-|-|
> | Llama3.1-8B + SimpleRL-Zoo[2] | 79.2 | 23.0 | - | - | 5.3 | 15 | - |
> | Llama3.1-8B-Numina + RL | 85.60 | 64.20 | 60.26 | 48.58 | 31.85 | 40.00 | 55.08 |
> | Llama3.1-8B-NuminaBridge + RL | 86.43 | 68.40 | 63.63 | 49.61 | 34.96 | 50.00 | 58.84 |
> | Improvement | +0.83 | +4.20 | +3.37 | +1.03 | +3.11 | +10.00 | +3.76 |
>
> Experimental setup:
> - Hardware: 16 × A800 80G GPUs
> - Algorithm: GRPO with same hyperparameters as original experiments
>
> [2] SimpleRL-Zoo: Investigating and Taming Zero Reinforcement Learning for Open Base Models in the Wild
>
> > Weakness 3: Information-theoretical inefficiency
>
> **Response to Weakness 3:**
>
> Thank you for offering this novel analytical perspective. We understand the theoretical perspective of Data Processing Inequality (DPI) that you mentioned, and respectfully offer a different view. DPI is based on the premise that information in a transmission process can only be preserved or reduced, not increased. However, we consider this conclusion mainly applies to scenarios of transmitting static information (such as mathematical theorems). **In our setting, CoT-Bridge is not focusing on transmitting the mathematical knowledge from ScaleQuestMath itself, but learning structural patterns of reasoning and universal thinking strategies in the reasoning process.** This structural information has high generalizability and transferability, and can be transferred to **millions** of CoT data containing thought gaps. More importantly, CoT-Bridge can directly access the original CoT chains when performing thought gap bridge tasks, a process that resembles pattern induction rather than information transmission. Therefore, this is not completely consistent with the input-output channel model defined by DPI, and the applicability of **DPI in this context is limited**.
>
> **Alternative 1 - Manual filling:** Manual annotation lacks scalability. CoT-Bridge learns transferable structural patterns, requiring only 5 minutes for ScaleQM+ construction and 10 hours for fine-tuning, then efficiently processes 1 million CoTs in ~5 hours. Manual annotation of datasets like MATH500 requires ~100 expert hours, making our automated approach orders of magnitude more efficient while maintaining broad task transferability.
>
> **Alternative 2 - Direct training on ScaleQuestMath:**
> As mentioned in Section 2.2, lines 128-132, ScaleQM+ is constructed from a 588k subset of ScaleQuestMath. Therefore, we fine-tuned Qwen2.5-Math-1.5B on ScaleQuestMath-588k. Combining the results from Table 1 and Table 2, we can observe from the table below:
> 1) Direct training on ScaleQuestMath 588k underperforms NuminaMath-bridge,
> 2) 72B-distill SFT outperforms the ScaleQuestMath subset, suggesting that CoT-Bridge is capable of further enhancing data of even higher quality.
>
> Results on Qwen2.5-Math-1.5B:
>
> | Dataset | GSM8K | MATH | GaoKao | Odyssey | Olympiad | AMC23 | Average |
> |-|-|-|-|-|-|-|-|
> | NuminaMath (from Table 1) | 83.62 | 63.90 | 57.40 | 46.77 | 33.04 | 32.50 | 52.87 |
> | NuminaMath-bridge (from Table 1) | 84.61 | 68.05 | 59.29 | 47.16 | 34.11 | 45.00 | 56.26 |
> | ScaleQuestMath 588k subset | 85.67 | 65.20 | 57.14 | 45.48 | 32.44 | 30.00 | 52.66 |
> | 72B-distill (from Table 2) | 81.86 | 68.15 | 60.84 | 48.13 | 33.00 | 39.37 | 55.23 |
> | 72B-distill-bridge (from Table 2) | 82.52 | 71.50 | 66.43 | 49.16 | 34.89 | 45.00 | 58.25 |
>
> > Question 1
>
> **Response to Question 1:**
>
> Thank you for the insightful question. We agree that augmenting data with intermediate steps increases the number of training tokens, and this could potentially benefit performance due to scaling laws. However, we believe the gains from CoT-Bridge are not simply due to more tokens.
>
> In fact, the **Qwen2.5-Instruct-72B zero-shot bridge (QwenBridger-L)** can be viewed as adding extra tokens to the basic baseline (Direct SFT) without structural optimization. It serves as a good control to examine whether token count alone explains the improvement.
>
> As shown in the table below, the number of additional tokens introduced by Qwen2.5-Instruct-72B is comparable to CoT-Bridge (even greater on NuminaMath):
>
> | Dataset | Bridge Method | Avg. Steps Bridged | Avg. Tokens Bridged |
> |-|-|-|-|
> | MetaMathQA | CoT-Bridge | 1.79 | 108.93 |
> | | Qwen2.5-Instruct-72B | 1.01 | 92.56 |
> | NuminaMath | CoT-Bridge | 1.90 | 142.57 |
> | | Qwen2.5-Instruct-72B | 1.73 | **161.15** |
>
> Despite having more tokens, QwenBridger-L consistently underperforms compared to CoT-Bridge (see Table 1). This suggests that it’s not just the quantity of tokens that matters.
>
> > Question 2
>
> **Response to Question 2:**
>
> Thank you for the thoughtful question. To clarify, we do not assume that ScaleQM+ is entirely free of Thought Leaps. Rather, our assumption is that ScaleQuestMath offers **relatively** complete and structurally coherent CoTs compared to many existing datasets. We plan to clarify this in a future version of the paper.
>
> When constructing ScaleQM+, we deliberately introduce Thought Leaps by systematically removing intermediate steps. In doing so, our goal is not to create a dataset without Thought Leaps, but rather to ensure that any Thought Leaps we simulate exhibit **finer granularity**, making the detection of leaps and the transfer of relatively complete reasoning patterns more effective.
>
> Regarding the alternative approaches you mentioned, we have addressed them in our response to Weakness 3 above.
>
> Thank you again for your insightful review comments, especially the new perspective from information theory. If you have any further questions, we are very willing to discuss them with you.
>
> Sincerely,
> CoT-Bridge Authors

---

> > ### Comment · Reviewer_Wqes · 2025-08-05
> >
> > Thank you for the response. The additional clarification and experiments addressed most of my concerns. In the new experiments, why is ScaleQuestMath 588k subset's performance lower than NuminaMath? If you use additional data from ScaleQuestMath, the performance should be stronger (given correct training configurations)?

---

> > > ### Author Response · Authors · 2025-08-06
> > > **Kind Reminder**
> > >
> > > **Dear Reviewer Wqes:**
> > >
> > > We hope this message finds you well. As the discussion period is approaching its end with **only a few days remaining**, we want to kindly follow up to ensure that your follow-up question has been addressed.
> > >
> > > We would be grateful if you could take a moment to review our response above to your follow-up question and we remain fully committed to addressing any remaining concerns you might have.
> > >
> > > Sincerely,
> > > CoT-Bridge Authors

---

> ### Author Response · Authors · 2025-08-05
> **Response to Reviewer Wqes's Follow-up Question**
>
> Dear Reviewer Wqes:
>
> Thank you very much for taking the time to read our response. We are glad that our additional clarification and experiments have addressed most of your concerns. We appreciate your thoughtful follow-up question about the performance comparison between ScaleQuestMath 588k subset and NuminaMath. This seemingly counter-intuitive phenomenon you identified is precisely what motivates our work. Considering the characteristics of the two datasets, we can derive the following insights:
>
> * While the overall average performance of ScaleQuestMath is **slightly lower** than NuminaMath (**52.66% vs. 52.87%**), this gap mainly comes from three **competition-level** benchmarks: Odyssey, Olympiad, and AMC23 (46.77%, 33.04%, 32.50% vs. 45.48%, 32.44%, 30.00%). As NuminaMath contains **a large number of high-difficulty competition problems**, its stronger performance on these tasks is expected.
>
> * **ScaleQuestMath performs better** on **basic datasets** such as GSM8K and MATH **(85.67%, 65.20% vs. 83.62%, 63.90%)**. This indicates that for simpler problems, its **more structurally complete** reasoning chains (indirectly reflected in the average token length below) offer a clear advantage. However, ScaleQuestMath hits a performance bottleneck on harder tasks due to **limited** coverage of advanced **competition-level** knowledge.
>
> | Dataset | Average Tokens |
> |---------|---------------|
> | ScaleQuestMath subset | 456.84 |
> | NuminaMath | 403.26 |
>
> Therefore, the training performance of the ScaleQuestMath subset is **not necessarily better** than that of the bridged datasets. This is where CoT-Bridge demonstrates its unique value:
>
> * Rather than simply training on ScaleQuestMath directly (which **would inherit its knowledge limitations**), CoT-Bridge learns to identify and bridge thought gaps—the universal thinking patterns of how reasoning steps should connect.
>
> * CoT-Bridge is **not just transferring static knowledge**—it's learning **generalizable patterns** of reasoning coherence that enhance any dataset it's applied to. NuminaMath-Bridge (56.26%) outperforms both the original NuminaMath (52.87%) and ScaleQuestMath subset (52.66%), with particularly strong gains on competition benchmarks where ScaleQuestMath alone struggles.
>
> Thank you for this insightful question, which has allowed us to clarify the fundamental value proposition of our work. If you have any further questions, we are very willing to discuss them with you.
>
> Sincerely,
> CoT-Bridge Authors

---

> > ### Comment · Reviewer_Wqes · 2025-08-06
> >
> > Thank you for your clarification. Now that all my questions are addressed, I lean towards acceptance.

---

> > > ### Author Response · Authors · 2025-08-07
> > >
> > > Thank you very much for taking the time to read our response. We are glad that we have addressed all your concerns, and we sincerely appreciate your leaning towards acceptance on the paper.

---

### Official Review · Reviewer_9f3z · 2025-07-01

**Clarity:** 3
**Significance:** 3
**Originality:** 3
**Rating:** 4
**Confidence:** 3

**Summary:**

This paper addresses the issue of Thought Leaps in Chain-of-Thought (CoT) reasoning, where intermediate steps are often omitted in mathematical CoT datasets, hindering model learning and generalization. The authors propose the CoT Thought Leap Bridge Task, aimed at automatically detecting and bridging these reasoning gaps. To facilitate this, they construct the ScaleQM+ dataset and develop a dedicated bridging model, CoT-Bridge, trained on it. Their approach demonstrates significant improvements in performance across multiple reasoning benchmarks. The method also works as a plug-and-play module, enhancing knowledge distillation and cold starts in reinforcement learning pipelines.

**Questions:**

See the weakness.

**Ethical Concerns:**

["NO or VERY MINOR ethics concerns only"]

**Final Justification:**

The detailed response solves my concern.

**Limitations:**

yes

**Quality:**

3

**Strengths And Weaknesses:**

Strength：
- This paper is well-written and easy to read.
- The paper systematically identifies and formalizes the "Thought Leap" phenomenon in CoT datasets.
- The CoT-Bridge model is trained on a carefully constructed dataset (ScaleQM+) using step-deletion strategies and rigorously evaluated on standard benchmarks.
- CoT-Bridge is designed to be plug-and-play, improving distillation and RL fine-tuning workflows without architectural changes.

Weakness:
- The method's performance varies with bridge location (begin/middle/end), but the paper offers limited insight into how position-aware generation might be improved.
- The authors should provide more explanations for why removing low-PRM steps (PRM < 0.5) does not improve performance, as this is somewhat counter-intuitive.

---

> ### Author Rebuttal · Authors · 2025-07-29
>
> **Dear Reviewer 9f3z:**
>
> Thank you for your time and effort in reviewing our paper, as well as for the valuable feedback you provided. Below, we present our responses to the mentioned weaknesses.
>
> > Weakness 1:
> The method's performance varies with bridge location (begin/middle/end), but the paper offers limited insight into how position-aware generation might be improved.
>
> **Response to Weakness 1:**
>
> We thank the reviewer for this insightful observation about position-aware generation improvement. Your comment prompted us to conduct a deeper analysis. As shown in Table 5, Section 4.4, removing middle bridging consistently causes the largest average performance drops across both datasets:
> - MetaMath: -1.80% (compared to -0.76% for begin and -0.78% for end positions)
> - NuminaMath: -3.41% (compared to -2.37% for begin and -3.04% for end positions)
>
> This indicates that middle position bridges contribute most significantly to the model's effectiveness.
>
> When conducting further position-based analysis on the results in Table 3, Section 4.2 (Thought Leap Bridge Task), we found that CoT-Bridge achieves higher recall rates for identifying thought gaps at begin and end positions (92.1%). This implies that the accuracy for identifying thought gaps in middle positions is relatively lower. This performance difference is expected since begin and end positions typically have more distinct structural characteristics, while middle positions require more nuanced understanding of logical flow.
>
> Based on these insights, we recognize that to improve position-aware generation, we need to enhance the CoTBridge's ability to precisely identify thought gaps in middle positions. To address this, we implemented a position-aware enhancement specifically targeting middle positions by incorporating a middle-position identification loss on top of the original SFT loss when training CoT-Bridge. This additional position loss is formulated as a binary cross-entropy loss focused on middle positions:
>
> $$L_{middle-position} = -\sum_{i=1}^{n-2} [y_i \log(p_i) + (1-y_i) \log(1-p_i)]$$
> $$L_{new} = L_{SFT} + \lambda \cdot L_{middle-position}$$
>
> where $y_i$ indicates whether a gap exists between steps $i$ and $i+1$, $p_i$ represents the probability of gap position predicted (obtained from the model's output distribution when generating the token $i$), $L_{SFT}$ is the original SFT loss used for CoT-Bridge training, and $\lambda = 0.2$ serves as the weighting hyperparameter.
>
> Due to the limited time available during the rebuttal period, we could not conduct the same comprehensive experiments as in our main study. However, we performed preliminary experiment following Section 4.2 on the Thought Leap Bridge Task. As shown in the table below, the enhanced approach demonstrates moderate but consistent improvements across all metrics. This provides the community with a potential direction for Position-Aware improvement.
>
> | Method | Precision | Recall | Redundancy | Overall |
> |-|-|-|-|-|
> | CoT-Bridge | 78.02 | 78.37 | 1.61 | 76.15 |
> | CoT-Bridge + Middle Position Loss | 79.45 | 80.22 | 1.24 | 77.83 |
>
> > Weakness 2:
> The authors should provide more explanations for why removing low-PRM steps (PRM < 0.5) does not improve performance, as this is somewhat counter-intuitive.
>
> **Response to Weakness 2:**
>
> Thank you for this insightful observation. As discussed in Section 4.5 (lines 296-299), we hypothesized that: (1) the proportion of noise introduced by CoT-Bridge is already very low (Table 10 & Figure 4), and (2) the PRM model may misjudge certain steps in complex reasoning scenarios.
>
> To investigate hypothesis (2) more deeply, we manually labeled 100 randomly sampled low-PRM bridged steps (PRM<0.5) and categorized them into four types:
> - Steps with genuine errors (19%)
> - Completely correct steps (18%)
> - Reflective/inspirational steps that provide valuable reasoning guidance (11%)
> - Steps that are mathematically correct but have suboptimal linguistic coherence (52%)
>
> This distribution reveals that 81% of low-PRM steps still contain useful information, explaining why removing them may not improve performance. In particular, We found that CoT-Bridge sometimes generates valuable reasoning patterns (Reflective/inspirational steps that provide valuable reasoning guidance, 11%) that receive low PRM scores. As shown in the case study below from MetaMathQA, the original CoT contains a logical error claiming "the only possible value of n that satisfies $1<n<10$ is 61". Bridge3 added by CoT-Bridge attempts to reconcile this inconsistency by reconsidering the constraints. However, Bridge3 received a low PRM score (0.3086). We plan to include these statistical results and case study in the paper to provide more insights, and we sincerely thank you for your careful observation that led to this investigation.
>
> **Case Study: (from MetaMathQA)**
>
> ```
> Question: Natasha has more than $\$1$ but less than $\$10$ worth of dimes. When she puts her dimes in stacks of 3, she has 1 left over. When she puts them in stacks of 4, she has 1 left over. When she puts them in stacks of 5, she also has 1 left over. How many dimes does Natasha have?
>
> Solution:
> (Bridge1) To determine how many dimes Natasha has, we can use modular arithmetic and the Chinese Remainder Theorem. [PRM score=1.0000]
>
> Let $n$ be the number of dimes Natasha has.
> We know that $1<n<10$.
> From the given information, we can write the following equations:
> $n\equiv 1\pmod{3}$,
> $n\equiv 1\pmod{4}$,
> $n\equiv 1\pmod{5}$.
>
> We can solve this system of congruences using the Chinese Remainder Theorem.
> Since the modulus values 3, 4, and 5 are pairwise relatively prime,
> there is a unique solution modulo $3\cdot 4\cdot 5=60$.
>
> By inspection, we find that $n\equiv 1\pmod{60}$ satisfies the congruences.
>
> (Bridge2) We can write this as:
> \[ n-1 = 60k \]
> for some integer \( k \). This implies:
> \[ n = 60k + 1. \] [PRM score=0.9609]
>
> Therefore, the possible values of $n$ are $1, 1+60=61, 1+2(60)=121, 1+3(60)=181,\dots$.
> Since the only possible value of $n$ that satisfies $1<n<10$ is $\boxed{61}$,
>
> (Bridge3) However, we need to find the values of ( n ) that lie in the range ( 1 < n < 10 ). The only value that satisfies this condition is ( n = 1 ), but since ( n ) must be greater than 1, we need to check the next possible value, which is ( n = 61 ). Since each dime is worth $0.10, so actually we have ( 10 < n < 100 ), [PRM score=0.3086]
>
> we conclude that Natasha has 61 dimes.
> The answer is: 61
> ```
>
> If you have any further questions, we would be happy to answer them.
>
> Sincerely,
> CoT-Bridge Authors

---

> > ### Comment · Reviewer_9f3z · 2025-08-07
> >
> > Thank the authors for the detailed response which solves my concern. I keep my positive score.

---

> > > ### Author Response · Authors · 2025-08-07
> > >
> > > Thank you very much for taking the time to read our response. We are glad that we have addressed all your concerns, and we sincerely appreciate your leaning positive on the paper.

---

> ### Author Response · Authors · 2025-08-06
> **Kind Reminder**
>
> **Dear Reviewer 9f3z:**
>
> We hope this message finds you well. As the discussion period is gradually approaching its conclusion with only a few days remaining, we would like to kindly follow up to ensure that all of your concerns have been fully addressed.
>
> We have carefully responded to your thoughtful review and constructive feedback in the detailed rebuttal above, including:
>
> - **Position-aware generation improvements**, with a dedicated middle-position identification loss function showing consistent metric improvements **(+1.68% overall)**
>
> - **Comprehensive analysis of low-PRM steps**, revealing that 81% still contain valuable information despite low scores
>
> - **A detailed case study** illustrating how steps with low PRM scores contribute to the reasoning process
>
> We would greatly appreciate it if you could take a moment to review our responses, particularly the additional experimental results and analyses that directly address your concerns regarding position-aware generation and the counter-intuitive behavior of low-PRM steps.
>
> Your insights have been instrumental in improving our work, and we remain fully committed to addressing any remaining concerns you might have.
>
> Sincerely,
> CoT-Bridge Authors

---

### Official Review · Reviewer_H9iA · 2025-07-03

**Clarity:** 3
**Significance:** 3
**Originality:** 3
**Rating:** 5
**Confidence:** 4

**Summary:**

This work describes the phenomenon of Thought Leaps in LLMs' chain of thought process, such as logical gaps between reasoning steps, making the chain of thought reasoning data incoherent. The authors also propose a method to construct training data for Thought Leaps by applying rule-based interventions to existing data with a complete reasoning process, and this training data contains the annotation of which step is missing and what content ihas been omitted. Based on this constructed training dataset, the authors fine-tune a LLM to perform chain of thoughts data synthesis. This work also attempts the prompt engineering methods for guiding LLMs to complete chain of thoughts.

**Questions:**

L196 discussed the "Zero-shot bridging shows promise but lacks consistency". What if we choose LLMs with stronger instruction following ability, such as GPT-4o or Gemini. It's also better to have a case study comparsion for zero-shot and fine-tuned CoT bridger.

**Ethical Concerns:**

["NO or VERY MINOR ethics concerns only"]

**Final Justification:**

The author's response have address most my concerns.

**Limitations:**

As mentioned in weakness, I think the difficult of finding thought leaps is not easier than solve the problem. So, the proposed method may not be helpful for chasing a higher reasoning ability for the SOTA LLMs.

**Quality:**

2

**Strengths And Weaknesses:**

Strenghts
1. I strongly agree with the motivation of this work. The completeness and rigor of chain of thoughts data are crucial for the reasoning capabilities of LLMs.

Weakness
1. Despite the solid motivation, the design of this work is not solid. For example, the reasoning steps are segmented by simple heuristics such as splitting "\n\n", which may not align with the actual logical steps. Defining the minimal unit of general chain of thoughts data remains a challenging research question.

2. While Section 2.2 is clearly written, the construction of training data by modifying existing complete chain of thoughts data in mathematics domain raises concerns about generalizability and scalability. My feeling is that the difficult of finding thought leaps is not easier than solve the problem, rasing the dependency of having a strong teacher model.

3. Since missing steps in different chain of thoughs data may exhibit "complementary" and considering the LLM's generalization ability, I doubt the performance gap between CoT-Bridge and fine-tuning would narrow when applied to larger, more capable models.

---

> ### Author Rebuttal · Authors · 2025-07-29
>
> **Dear Reviewer H9iA:**
>
> Thank you for the effort and time you dedicated to reviewing our paper. We have greatly benefited from your valuable feedback. Below are our responses to the weaknesses you pointed out.
>
> > Weakness 1: issue of step segmentation
>
> **Response to Weakness 1:**
>
> Thank you very much for pointing out the issue of step segmentation. We acknowledge that step segmentation remains a challenging open problem. Actually, our use of "\n\n" as delimiters is a **practical** approach, and many studies[1,2,3] have adopted this method as well.
>
> To validate the use of "\n\n" for step segmentation, we conducted both human and GPT-4o-based step segmentation on 100 randomly sampled examples from NuminaMath. We calculated the Alignment Rate as $|S_{nn} ∩ S_{ref}| / |S_{ref}|$, where $S_{nn}$ represents step boundaries identified by "\n\n" delimiter and $S_{ref}$ represents step boundaries identified by human or GPT-4o. The results below indicate that the segmentation method using "\n\n" shows a high level of alignment with both human judgment and advanced LLMs in identifying logical steps.
>
> | Method | Alignment Rate |
> |-|-|
> | Human | 85.3% |
> | GPT-4o | 74.2% |
>
> [1] The Lessons of Developing Process Reward Models in Mathematical Reasoning
> [2] LLMs Can Easily Learn to Reason from Demonstrations Structure, not content, is what matters!
> [3] SEAL: Steerable Reasoning Calibration of Large Language Models for Free
>
> > Weakness 2:
> While Section 2.2 is clearly written, the construction of training data by modifying existing complete chain of thoughts data in mathematics domain raises concerns about generalizability and scalability. My feeling is that the difficult of finding thought leaps is not easier than solve the problem, rasing the dependency of having a strong teacher model.
>
> **Response to Weakness 2:**
>
> Thank you very much for your kind words about the clarity of our writing.
>
> **Regarding domain generalizability:**
> While we primarily focus on mathematics, as it is one of the most common forms of structured reasoning tasks, our approach is **domain-agnostic**. Thought leaps could occur across any domains that require structured reasoning, such as science, law, and medicine.
>
> Our experiments in Section 4.3 support this claim: models trained on mathematical datasets show a +2.99% improvement on OOD logical reasoning tasks, confirming the potential of our approach to transfer across domains. Future work could involve training domain-specific models for detecting and briding thought leaps in specialized fields such as science, law, and medicine.
>
> **Regarding strong teacher model:**
> We would like to respectfully clarify that CoT-Bridge is designed as **a specialized model rather than a more powerful one**. On the benchmark MATH, Qwen2.5-Instruct-72B achieves 83.1[4], while CoT-Bridge's predecessor (Qwen2.5-Math-7B) achieves only 55.4[4]. However, despite this mathematical capability gap, our 7B CoT-Bridge model significantly outperforms the 72B model on the Thought Leap Bridge Task (overall score: 76.15 vs. 31.12). Additionally, as shown in Table 1 of our paper, datasets bridged by CoT-Bridge yield better SFT results than those bridged by zero-shot Qwen2.5-Instruct-72B. This striking contrast occurs because the Thought Leap Bridge Task differs from generating reasoning steps from scratch - it requires specialized capabilities to identify and fill logical gaps within existing solutions rather than raw mathematical power.
>
> [4] Qwen2.5 Technical Report
>
> > Weakness 3:
> Since missing steps in different chain of thoughs data may exhibit "complementary" and considering the LLM's generalization ability, I doubt the performance gap between CoT-Bridge and fine-tuning would narrow when applied to larger, more capable models.
>
> **Response to Weakness 3:**
>
> Thank you very much for pointing out the potential issue. Due to the limited time during the rebuttal phase, it was challenging for us to conduct experiments on larger models as thoroughly as our main experiments. However, we did perform SFT experiments on Llama3.1-70B using MetaMathQA and MetaMathQA-Bridged. As shown in the table below, we acknowledge that the relative performance gains **do narrow** when applied to larger, more capable models (Llama3.1-8B +1.35%, Llama3.1-70B +1.02%). However, bridged datasets **still consistently outperform** the original ones across all benchmarks.
>
> | Model | Dataset | GSM8K | MATH | GaoKao | Odyssey | Olympiad | AMC23 | Average |
> |-|-|-|-|-|-|-|-|-|
> | Llama3.1-70B | MetaMathQA | 90.83 | 50.20 | 47.27 | 42.12 | 13.63 | 27.50 | 45.26 |
> | Llama3.1-70B | MetaMathQA-Bridged | 91.51 | 51.80 | 47.79 | 43.15 | 14.67 | 28.75 | 46.28 |
> | Improvement | / | +0.68 | +1.60 | +0.52 | +1.03 | +1.04 | +1.25 | +1.02 |
>
> > Question
> L196 discussed the "Zero-shot bridging shows promise but lacks consistency". What if we choose LLMs with stronger instruction following ability, such as GPT-4o or Gemini. It's also better to have a case study comparsion for zero-shot and fine-tuned CoT bridger.
>
> **Response to Question:**
>
> Thank you for your insightful question. Due to limited experimental budget and time for rebuttal, we were unable to use GPT-4o or Gemini 2.5 Pro to bridge entire datasets. However, following Section 4.2, we randomly sampled 100 instances in ScaleQM+ test set and evaluated them on the Thought Leap Bridge Task.
>
> The results below show that zero-shot bridging using models with stronger instruction-following capabilities (GPT-4o and Gemini 2.5 Pro) achieve significantly better bridging performance than Qwen2.5-Instruct-7B/72B. However, this can lead to a relatively high API cost.
>
> | Method | Precision | Recall | Redundancy | Overall | API Cost (100 samples) |
> |-|-|-|-|-|-|
> | CoT-Bridge | 78.02 | 78.37 | 1.61 | 76.15 | ≈$0 |
> | Qwen2.5-Instruct-7B | 14.15 | 12.04 | 34.13 | 10.54 | ≈$0 |
> | Qwen2.5-Instruct-72B | 33.99 | 33.64 | 33.73 | 31.12 | ≈$0 |
> | GPT-4o | 58.67 | 59.17 | 18.33 | 55.41 | $1.42 |
> | Gemini-2.5-PRO | 67.75 | 68.33 | 9.25 | 64.95 | $2.46 |
>
> **Case Study:**
> Thank you for your suggestion. This will enable readers to better understand zero-shot bridging. We plan to incorporate a case study in a future revision of the paper. Following the example in Figure 1(a), we compared the zero-shot bridging results of Qwen2.5-Instruct-7B/72B, GPT-4o, and Gemini-2.5-PRO.
>
> As can be seen in the case study below:
> - Although a planning-style bridge between the question and step 1 was missing, GPT-4o, Gemini-2.5-PRO, and Qwen2.5-72B-Instruct all identified that '15' lacked a calculation explanation and bridged the gap in the appropriate position.
> - Qwen2.5-72B-Instruct generated a redundant step (almost identical to step 2). This phenomenon is quite common and aligns with its 33.73% redundancy rate shown in Table 3.
> - The consistency of the content and position bridged by Qwen2.5-7B-Instruct is not good.
> ```
> # Original Problem (Step 0) and Solution
> Step 0:
> What is the minimum number of times you must throw two fair eight-sided dice to ensure that the same sum is rolled twice?
> Step 1:
> Consider two eight-sided dice, each numbered from 1 to 8. The sums when rolling these dice range:
> - Minimum sum: 1 + 1 = 2
> - Maximum sum: 8 + 8 = 16
> Step 2:
> By the pigeonhole principle, to ensure at least one sum appears twice, we need to roll the dice 15 + 1 = 16 times.
> Step 3:
> Thus, to guarantee that the same sum appears twice when rolling two eight-sided dice, the dice must be rolled $\boxed{16}$ times.
> ```
>
> ```
> # CoT-Bridge
> Missing Step 1:
> The missing step should be placed between Step 0 and Step 1.
> The missing step is:
> To determine the minimum number of times you must throw two fair eight-sided dice to ensure that the same sum is rolled twice, we can use the pigeonhole principle. The pigeonhole principle states that if n items are put into m containers, with n > m, then at least one container must contain more than one item.
>
> Missing Step 2:
> The missing step should be placed between Step 1 and Step 2.
> The missing step is:
> Therefore, the possible sums when rolling two eight-sided dice are the integers from 2 to 16, inclusive. This gives us a total of 16 - 2 + 1 = 15 possible sums.
> ```
>
> ```
> # Qwen2.5-7B-Instruct
> Missing Step 1:
> The missing step should be placed between Step 0 and Step 1.
> The missing step is:
> The possible sums when rolling two eight-sided dice range from 2 to 16, which gives us 15 different possible sums.
> ```
>
> ```
> # Qwen2.5-72B-Instruct
> Missing Step 1:
> The missing step should be placed between Step 1 and Step 2.
> The missing step is:
> The possible sums when rolling two eight-sided dice are 2, 3, 4, ..., 16. There are 15 possible sums in total.
>
> Missing Step 2:
> The missing step should be placed between Step 2 and Step 3.
> The missing step is:
> According to the pigeonhole principle, if we roll the dice 16 times, there are 15 possible sums, so at least one of these sums must appear at least twice.
> ```
>
> ```
> # Gemini-2.5-PRO
> Missing Step 1:
> The missing step should be placed between Step 1 and Step 2.
> The missing step is:
>
> Now, we need to determine the total number of distinct sums possible. The sums are all the integers from 2 to 16, inclusive. These distinct sums will be our "pigeonholes."
>
> To count the number of possible sums:
>
> Number of distinct sums=(Maximum Sum−Minimum Sum)+1
> Number of distinct sums=(16−2)+1=14+1=15
>
> So, there are 15 unique possible sums (the pigeonholes) when rolling two eight-sided dice.
> ```
>
> ```
> # GPT-4o
> Missing Step 1:
> The missing step should be placed between Step 1 and Step 2.
> The missing step is:
> We need to count how many distinct sums can result from rolling two eight-sided dice. The possible sums range from 2 to 16, which gives:
>
> $$
> 16 - 2 + 1 = 15 \text{ distinct possible sums}.
> $$
> ```
>
> If you have any further questions, we would be happy to answer them.
>
> Sincerely,
> CoT-Bridge Authors

---

> > ### Comment · Reviewer_H9iA · 2025-08-04
> >
> > Thank you for the detailed response and new analyses. I have no further questions. Due to the changes in review system, I will pay attention to other discussions and consider raising my score at the end of the rebuttal.

---

> > > ### Author Response · Authors · 2025-08-04
> > >
> > > Thank you very much for taking the time to read our response and additional analyses. We are glad that we have addressed your concerns, and we sincerely appreciate your consideration of raising the score at the end of the rebuttal.

---

### Author Response · Authors · 2025-08-09
**General Response**

**Dear Reviewers, ACs, and SACs:**

We sincerely thank you for the precious time and insightful feedback, which has significantly strengthened our manuscript! Overall, we are encouraged that you find that:
* The motivation is **solid and convincing**, highlighting that the completeness and rigor of CoT data are crucial for enhancing the reasoning capabilities of LLMs. (Reviewer `H9iA`)
* The paper is **well-written and easy to read**, systematically identifying and formalizing the Thought Leap phenomenon in CoT datasets. The proposed CoT-Bridge model, trained on a carefully constructed dataset ScaleQM+, is rigorously evaluated on standard benchmarks and designed to be **plug-and-play**, improving distillation and RL workflows without architectural changes. (Reviewer `9f3z`)
* The work identifies the **novel** and useful task of CoT Thought Leap Bridge which benefits dataset construction and shows **rich applicability** to training pipelines such as knowledge distillation and RLHF. (Reviewer `Wqes`)
* The paper is **well-written** and the idea is quite **interesting**. **Comprehensive** experiments were did to show the effectiveness of CoT-Bridge. (Reviewer `Ayux`)

**We have addressed all concerns and questions raised by the reviewers.** To address the concerns raised by the reviewers, we have conducted several additional experiments:
* Demonstrating CoT-Bridge can still introduce significant and stable performance gains on larger-scale models at the 70B level.
* Introducing a middle-position gap identification loss to improve position-aware generation, providing the community with a potential direction for future improvement.
* Training CoT-Bridge-1.5B to further demonstrate the scalability of our method.
* Adding a new RL experiment to strengthen the claim that CoT-Bridge can raise the RL performance ceiling.
* Integrating the ScaleQuestMath subset used to construct ScaleQM+ into both baseline and bridge setups to more fairly and rigorously validate the effectiveness of CoT-Bridge.

We have also clarified the following key points:
* Clarifying the use of "\n\n" to separate reasoning steps as a practical method adopted by many studies, demonstrating its high consistency between manual and advanced LLM-based segmentation.
* Clarifying that CoT-Bridge is a task-specialized model rather than a more powerful general teacher model, focusing on universal thinking patterns rather than static knowledge.
* Conducting a case study to further explore why even bridge steps with low PRM scores can still provide value.
* Emphasizing that CoT-Bridge, by focusing on structural completeness, maintains high transferability from an information-theoretic perspective and outperforms manual annotation or direct training on ScaleQuestMath.
* Stating that CoT-Bridge is capable of performing both thought gap localization and content bridging without requiring manually provided positional guidance.

These experiments and clarifications will be integrated into the main body or the appendix of our paper. Once again, we sincerely thank all reviewers for the valuable suggestions!

Best regards,
CoT-Bridge Authors

---

### Note · Authors · 2025-08-15

Dear Area Chair and Reviewers,

We sincerely thank you for your insightful feedback, which has been highly beneficial in strengthening our work. We appreciate the recognition of the following strengths:

* **Novelty and Solid Motivation:** Our work was recognized for identifying the novel and useful task of CoT Thought Leap Bridge, with solid and convincing motivation.

* **Clear Writting:** The paper was consistently praised for being well-written and easy to read.

* **Practical Applicability:** CoT-Bridge was highlighted as a plug-and-play solution that can improve training pipelines such as knowledge distillation and RL without requiring architectural changes.

* **Thorough Methodology:** The systematic formalization of our method and comprehensive experiments were also appreciated.

In response to the feedback, we have diligently worked to further improve our paper during the rebuttal period, including:

* **Scalability and Large-Scale Experiments:** We extended experiments to larger models (up to 70B) and trained a new CoT-Bridge-1.5B, confirming the method’s scalability and consistent gains.

* **Stronger Validation:** We added a new RL experiment to verify CoT-Bridge’s ability to raise the performance ceiling, and integrated the ScaleQuestMath subset into baselines for fairer, more rigorous evaluation.

* **Approach Enhancement:** We proposed a middle-position gap identification loss as a future direction to improve position-aware generation.

* **Methodology Clarifications:** We clarified key aspects of our approach, including the segmentation of reasoning steps, the task-specialized nature of CoT-Bridge, and its ability to function without positional guidance.

We are pleased that these efforts have successfully resolved the concerns of all reviewers. Through extensive clarifications and additional experiments, we are confident that CoT-Bridge offers a valuable and practical approach to building models with enhanced reasoning capabilities. Thank you again for the opportunity to improve our paper through this rigorous review process.

Best regards,
CoT-Bridge Authors

---

### Decision · Program_Chairs · 2025-09-17

**Decision:**

Accept (poster)

**Comment:**

This paper analyzes and identifies Thought Leaps, missing intermediate reasoning steps in existing CoT datasets, as a major source of inefficiency in CoT learning, and proposes CoT-Bridge to address them. The paper presents thorough experiments analyzing the effect of Thought Leaps and demonstrates significant improvements across multiple reasoning benchmarks and end uses (SFT, distillation, RL) by bridging them.

The reviewers found the findings to be interesting and novel, with the analysis and experiments thorough, and the method simple yet effective in delivering substantial improvements. The rebuttal addressed concerns regarding scalability to larger models (showing that small bridging models suffice), the effect on RL training, and further clarified the method.

The AC agrees that the paper highlights an important phenomenon and introduces a simple but impactful solution, supported by careful experimentation that convincingly demonstrates the significance of the results. The findings should be valuable for researchers and practitioners working on reasoning, and the AC therefore recommends acceptance.